# Regulating adsorption selectivity by charge-polarized $Au^{\delta-}$-$Cu^{\delta+}$ site for stable glucose electrooxidation

Yunpeng Liu [1], Xiaolong Tao[1], Chuqiang Huang[1], Kai Zhao[1], Binglu Deng [1] ✉ & Feng Peng [2] ✉

Electro-reforming of biomass into value-added chemicals offers a sustainable approach for future energy developments. However, noble metal catalysts toward glucose electrooxidation suffer from deactivation, poor selectivity, and limited power density. Here, we present $Au^{\delta-}$-$Cu^{\delta+}$ sites in AuCu alloy that serve as stable and efficient catalyst for selective glucose electrooxidation to potassium gluconate at high current density. AuCu alloy ensures the co-adsorption of $OH^-$ on electron-deficient $Cu^{\delta+}$ sites and glucose on electron-rich $Au^{\delta-}$ sites, stimulating the formation of oxidative *OH and intermediates. Selective adsorption of OH species on $Cu^{\delta+}$ sites also restrains the Au-OH formation and its subsequent oxidation to $AuO_x$, thereby preventing catalyst deactivation. Especially, for glucose electrooxidation, $Au_4Cu_2$ alloy delivers a high selectivity toward potassium gluconate (97.15%), along with a low potential of 0.74 V (versus reversible hydrogen electrode) to achieve industrial current density of 500 mA cm$^{-2}$. Furthermore, $Au_4Cu_2$ alloy realizes a stable electrolysis with a potassium gluconate productivity of 9.46 mmol cm$^{-2}$ h$^{-1}$ and Faraday efficiency of 93.60% in the membrane-free flow electrolyzer.

The prolonged reliance on fossil fuels has triggered substantial efforts in effective energy conversion technology. Biomass is considered as a $CO_2$-neutral, extensive, and renewable resource substitute to fossil fuels. Therefore, electrooxidation of biomass represents an attractive strategy for the sustainable energy economy, leveraging renewable electricity to convert biomass-based feedstocks into alternative fuels and chemicals[1-4]. Notably, electrocatalytic glucose oxidation (GOR) with a low overpotential stands out as a particularly promising option for large-scale impact[5-8]. Serving as a repeating unit of cellulose, glucose is the most abundant monosaccharide, which can be electrochemical converted to various value-added organic acids. Potassium gluconate (PGA), a product of alkaline GOR, proves highly applicable in pharmaceutical industry, food additives, agriculture, and other fields[9,10]. More importantly, GOR is thermodynamically and kinetically more favorable than oxygen evolution reaction (OER), which is identified as an alternative half reaction for enhancement of hydrogen

evolution reaction (HER) in alkaline water electrolysis[11-14]. This coupling strategy of GOR and HER highlights the potential for coproducing valuable PGA and $H_2$ fuel from renewable resources. As a result, arising from the expectation to enhance sustainability, the upcycling of glucose has generated significant research interest.

Transition metal-based catalysts such as Fe, Co, and Ni usually undergo structural reconstruction at high potentials during electrooxidation, leading to the formation of $M^{3+}$-$(OH)_{ads}$ with electrophilicity and oxidizability, which can unavoidably result in C−C bond breakage and production of low-value C1 chemicals (e.g., formic acid)[15-18]. Besides, noble metal catalysts (Au, Pt, and Pd) were also widely used for biomass molecules electrooxidation[19-22]. Previous studies have confirmed that the C-C bond cleavage behavior of noble metal catalysts exhibits significant differences: Pt and Pd are prone to causing deep C-C bond cleavage without selectivity; While Au catalysts, due to their moderate adsorption energy for intermediates, show excellent

[1]School of Materials and Energy, Foshan University, Foshan, China. [2]School of Chemistry and Chemical Engineering, Guangzhou University, Guangzhou, China. ✉e-mail: dengbl@fosu.edu.cn; fpeng@gzhu.edu.cn

C-C bond retention selectivity at low potential[23]. This characteristic makes Au an ideal catalyst for the electro-oxidation of biomass to produce high-value products. Glucose electrooxidation on Au proceeds via two potential-dependent pathways: (1) At <0.4 V vs. reversible hydrogen electrode (RHE), aldehyde group undergoes direct dehydrogenation to adsorption intermediates through OH-independent electron transfer; (2) Above 0.4 V vs. RHE, OH⁻ adsorption activates C=O bonds and accelerates dehydrogenation, promoting OH⁻ coupling to gluconic acid[23,24]. However, the electrooxidation activity of Au catalyst experiences rapid decay until deactivation at high potential[25,26]. For instance, applying an oxidation potential will result in the oxidation of Au-OH species into $AuO_x$, hindering the hydroxyl adsorption and the subsequent generation of oxidative hydroxyl adspecies (*OH)[27]. Therefore, controllable OH⁻ and glucose adsorption capacity and configuration play a pivotal role in selective GOR performance[28]. These present a significant challenge to develop noble metal catalysts toward GOR operated at high current density. Recently, an asymmetric pulse potential strategy was presented to resolve the stability of noble metal catalysts[29,30], by periodically applying a reductive potential for reduction of oxide on catalysts surface. Nevertheless, such an approach cannot fundamentally address the catalyst deactivation, while also causing an unnecessary energy waste.

Herein, we introduce charge-polarized $Au^{\delta-}$-$Cu^{\delta+}$ site into AuCu alloy via an electronegativity-driven charge redistribution strategy, and investigate its glucose electrooxidation performance. Specifically, $Au_4Cu_2$ alloy requires a GOR potential as low as 0.37 V vs. RHE to achieve the current density of 100 mA cm⁻², along with the high potassium gluconate yield and selectivity of up to 93.47% and 97.15%, respectively. Moreover, $Au_4Cu_2$ alloy realizes robust durability of GOR with steady Faraday efficiency (FE) (95.44−99.99%) over 120 h of electrolysis cycles. In situ electrochemical characterizations and theoretical calculations uncover the glucose and OH species co-adsorption capacity and conversion behaviors on alloy. The strong charge polarization of alloy enables OH⁻ absorbed on electron-deficient $Cu^{\delta+}$ sites and glucose absorbed on electron-rich $Au^{\delta-}$ sites, facilitating the formation of oxidative *OH and *CO-R carbonyl intermediates, thereby synergistically promoting GOR kinetics. And the favorable adsorption of OH species on $Cu^{\delta+}$ sites effectively suppresses the Au-OH formation and its further oxidation to $AuO_x$, ensuring stable glucose electrooxidation. Most encouragingly, a membrane-free flow electrolyzer equipped with $Au_4Cu_2$ alloy is designed for continuous GOR coupled with $H_2$ production. This membrane-free electrolysis system achieves PGA productivity of 9.46 mmol cm⁻² h⁻¹ with FE of 93.60%, demonstrating its capability for industrial application.

## Results

### Structural analysis of materials

AuCu alloy supported on Ni foam (NF) was synthesized via a simple electrodeposition method. Varied Au/Cu ratios in alloy were achieved via changing the proportion of precursors (denoted as $Au_xCu_y$). Scanning and transmission electron microscopy (SEM and TEM) images indicate that variations in Cu content of alloys induce a morphological change, undergoing an evolution from typical nanodendrite structure of Au metal to nanoparticle structure of $Au_4Cu_2$ alloy (Fig. 1a and Supplementary Figs. 1−3). Figure 1b depicts the high-resolution TEM (HRTEM) characterization of $Au_4Cu_2$, showing a crystalline lattice. The inset reveals the corresponding fast Fourier transform (FFT) viewed along the [011] zone axis. The lattice plane directions of ($1\bar{1}\bar{1}$), ($\bar{2}00$), and ($1\bar{1}1$) of $Au_4Cu_2$ alloy are corresponding to the face centered cubic (fcc) crystal structure. Figure 1c is the magnified lattice image of marked region in Fig. 1b, displaying a highly ordered structure with a spacing of 0.221 nm. $Au_4Cu_2$ alloy exhibits a slightly smaller lattice spacing than that of pure Au (0.237 nm) (Supplementary Fig. 4). This phenomenon implies the compressive strain in gold structure, caused

by the alloying with small copper atoms. Furthermore, the fcc lattice feature of ($\bar{1}11$), ($\bar{2}00$), and ($1\bar{1}1$) are consistent with the crystalline lattice with 57° (Fig. 1c). The increased interplanar angle from the standard angle of 54.7° may also be attributed to the lattice strain[31].

High-angle annular dark-field scanning transmission electron microscopy (HAADF-STEM) image enables the direct observation of atomic-scale structural details within $Au_4Cu_2$ alloy (Fig. 1d), which reveals a close-packed arrangement of atoms (Supplementary Fig. 5). More critically, intensity line profiles demonstrate the random insertion of Cu atoms within $Au_4Cu_2$ alloy (Fig. 1e), since the HAADF intensity is proportional to the square of the atomic number[32]. Energy-dispersive X-ray (EDX) elemental mapping images further confirm the uniform distribution of Au and Cu atoms in $Au_4Cu_2$ alloy (Fig. 1f). X-ray diffraction (XRD) patterns were performed to explore the impact of Cu introduction on AuCu alloy crystal (Fig. 1g). Apart from the diffraction peaks of Ni foam substrate (PDF#04-0850), pure Au shows the peaks at 38.18, 64.58, 77.55 and 81.72°, which correspond to the (111), (220), (311), and (222) planes of standard fcc Au (PDF#04-0784), respectively[33]. Upon alloying Cu with Au, the diffraction peaks of $Au_4Cu_2$ shift obviously to a higher angle, which is ascribed to the lattice compression[34]. Furthermore, the diffraction peaks shift to higher angle with an increasing amount of Cu (Supplementary Fig. 6), implying the formation of AuCu alloy. The compositions of the alloys derived from the inductively-coupled plasma-optical emission spectrometry (ICP-OES) and Vegard's law are listed in Supplementary Table 1. The concentration dependence of the lattice constant for AuCu alloys exhibits a positive deviation from the linear dependence calculated by Vegard's law (Supplementary Fig. 7). By simple arithmetic, we conclude that 5.85, 15.16, and 21.51% of the Cu remain out of the solid solution alloys in $Au_5Cu_1$, $Au_4Cu_2$, and $Au_3Cu_3$, respectively[35].

X-ray photoelectron spectroscopy (XPS) was conducted to analyze the surface structure and chemical states of $Au_4Cu_2$ alloy. Au exists in the metallic state in $Au_4Cu_2$ alloy, as confirmed by Au $4f_{7/2}$ peak at a binding energy of 84.4 eV assigned to metallic gold (Fig. 1h)[36]. Cu $2p$ spectrum of $Au_4Cu_2$ shows the similar peaks with that of pure Cu, and the surface of both contains metallic $Cu^0$ (952.0 and 932.2 eV) and $Cu^{2+}$ species (954.6 and 934.5 eV) (Fig. 1i)[8]. The $Cu^{2+}$ oxidation species occurs as a result of exposure to the atmosphere[37]. Compared with pure Au, the binding energies of $Au^0$ $4f$ peaks in $Au_4Cu_2$ negatively shift by about 0.3 eV, while the $Cu^0$ $2p$ peaks positively shift by about 0.6 eV, evidencing that Cu donates electrons to Au. Such electron transfer arises from electronegativity differences between Au (2.54) and Cu (1.90). Electron localization function (ELF) was performed to further evaluate the electronic interactions between Au and Cu atoms in alloy (Fig. 1j). For $Au_4Cu_2$ alloy, Au sites show higher degree of electron localization, and the electrons of Cu sites are more delocalized. Such electron redistribution disappears in pure Au (Fig. 1k). The electron transfer from Cu to its surrounding Au atoms was further demonstrated by differential charge density (Fig. 1l and Supplementary Fig. 8). It follows that introducing Cu atoms can donate abundant electrons to adjacent Au atoms, resulting in the formation of charge-polarized $Au^{\delta-}$-$Cu^{\delta+}$ sites. The charge polarization of $Au^{\delta-}$-$Cu^{\delta+}$ contributes positively to both catalytic GOR activity and durability of $Au_4Cu_2$ alloy.

### Electrocatalytic performances for glucose oxidation

The electrochemical performance of the electrocatalyst toward GOR was evaluated using a three-electrode configuration in 1 M KOH with 0.2 M glucose addition. Linear sweep voltammetry (LSV) curves were carried out under vigorous stirring to accelerate the mass transfer process. Figure 2a shows that $Au_4Cu_2$ alloy achieves the current density of 10 mA cm⁻² at only 0.14 V vs. RHE. In addition, $Au_4Cu_2$ delivers the current density of 500 mA cm⁻² at 0.74 V vs. RHE, distinctly outperforming the control samples of $Au_5Cu_1$ (355 mA cm⁻²), $Au_3Cu_3$ (69 mA cm⁻²), and Au (317 mA cm⁻²), respectively. Compared to

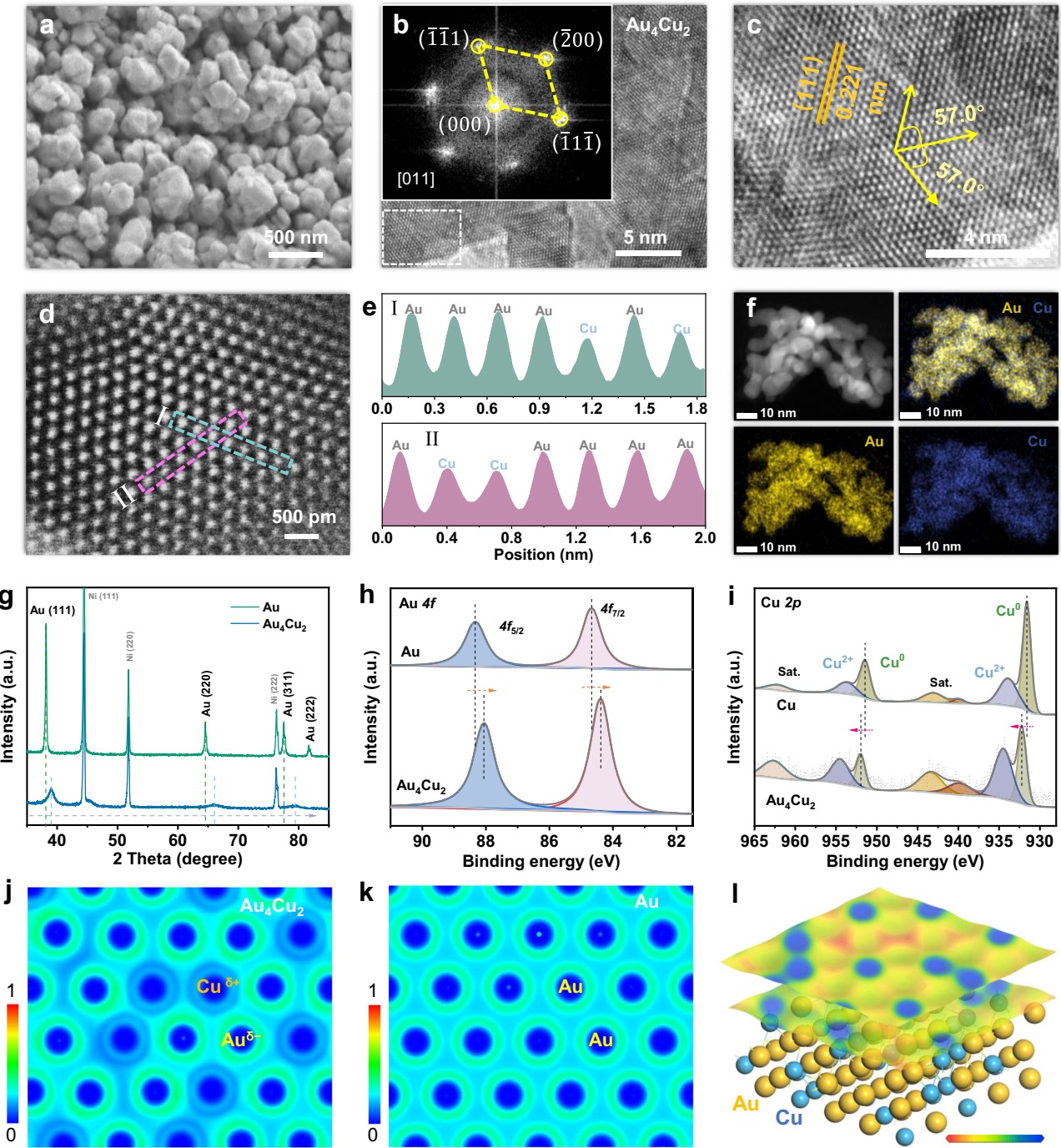

**Fig. 1 | Structural characterization of the $Au_4Cu_2$ alloy with $Au^{\delta-}$-$Cu^{\delta+}$ sites.** **a** SEM image of $Au_4Cu_2$. **b** HRTEM image and the corresponding FFT pattern of $Au_4Cu_2$. **c** HRTEM image taken from the marked region of (**b**). **d** Atomic-resolution HAADF-STEM image of $Au_4Cu_2$. **e** Intensity profiles corresponding to the labeled lines in (**d**). **f** EDX elemental mapping images of $Au_4Cu_2$. **g** XRD patterns. XPS spectra of (**h**) Au $4f$ and (**i**) Cu $2p$. ELF of (**j**) $Au_4Cu_2$ and (**k**) Au, and **l** differential charge density of $Au_4Cu_2$, where red and blue represent electrons being localized and electrons being delocalized, respectively. Source data for this figure are provided as a Source data file.

recently reported catalysts for biomass electrochemical reforming (Supplementary Table 2), the $Au_4Cu_2$ alloy in this work exhibits low overpotential, highlighting the competitive catalytic activity of $Au_4Cu_2$. Besides, Cu electrode shows a much poorer catalytic activity toward GOR compared with $Au_4Cu_2$ alloy electrode (Supplementary Fig. 9), which has a high initial potential of 1.31 V. The electrochemical surface area (ECSA) was carried out through cyclic voltammetry (CV) with different scan rates (Supplementary Fig. 10). Double-layer capacitance ($C_{dl}$) of activated $Au_4Cu_2$ is measured to be 6.64 mF cm$^{-2}$, surpassing 4.94, 5.65 and 5.79 mF cm$^{-2}$ of Au, $Au_5Cu_1$, and $Au_3Cu_3$,

respectively (Supplementary Fig. 11). To accurately assess the intrinsic activity of GOR, quasi-steady-state linear sweep voltammetry (QS-LSV) normalized through the ECSA is conducted based on the equilibrated current density under a given potential (Fig. 2b), where $Au_4Cu_2$ demonstrates the best catalytic activity of GOR. A steady state Tafel slope analysis was further applied to assess the reaction kinetics during the GOR, and the current density was also normalized through the ECSA (inset of Fig. 2b). The Tafel plots of $Au_4Cu_2$ is 101.4 mV dec$^{-1}$, smaller than that of $Au_5Cu_1$ (131.4 mV dec$^{-1}$), $Au_3Cu_3$ (220.8 mV dec$^{-1}$), and Au (169.1 mV dec$^{-1}$), revealing the faster GOR kinetics[38].

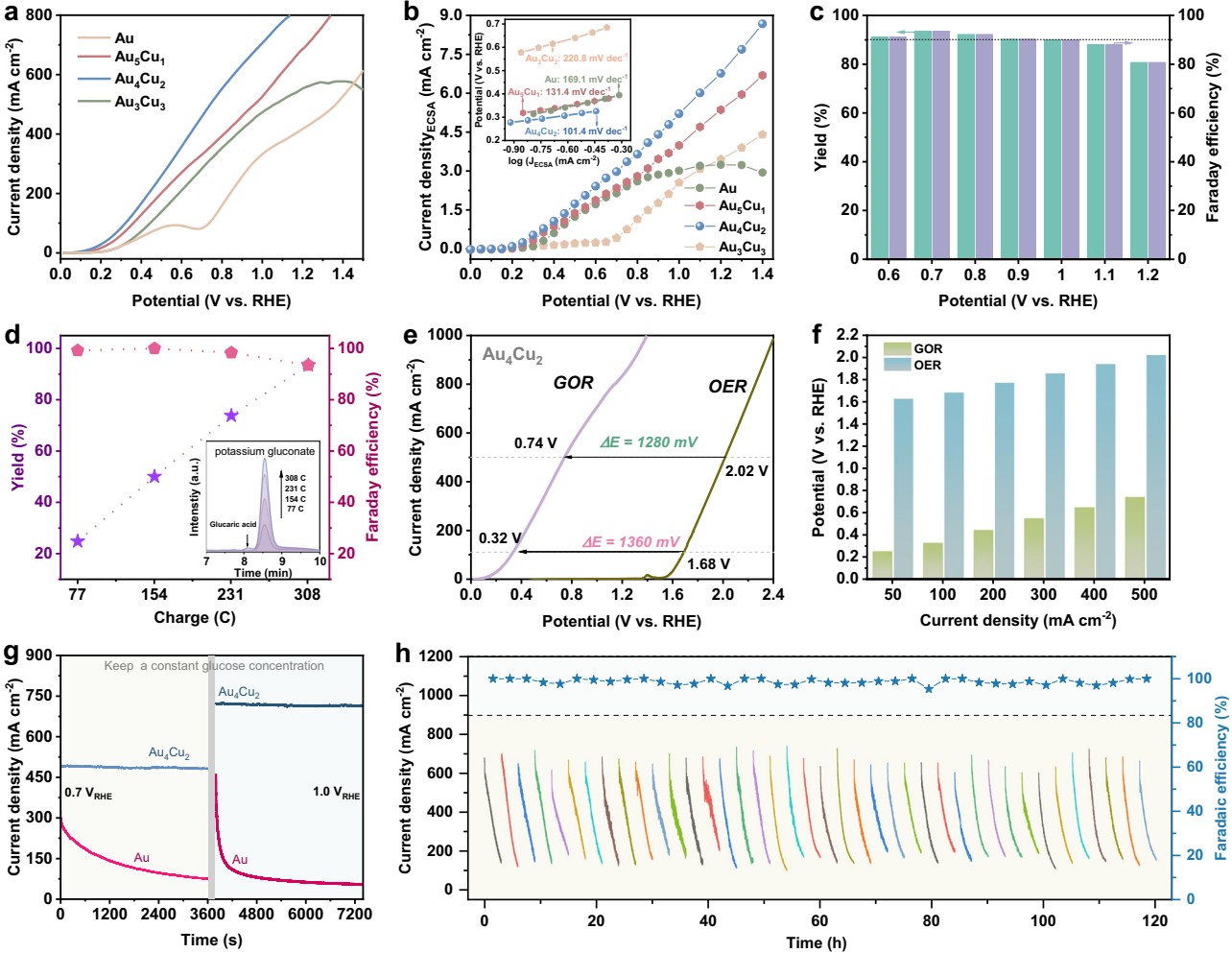

**Fig. 2 | Electrocatalytic GOR performance. a** LSV profiles of electrocatalysts for glucose oxidation in 1.0 M KOH and 0.2 M glucose. **b** ECSA-normalized QS-LSV plots toward GOR, inset is the steady state Tafel slopes. **c** Yields and FEs of $Au_4Cu_2$ toward GOR at different potentials. **d** PGA yields and FEs of $Au_4Cu_2$ under different passing charges at 0.7 V vs. RHE. **e** LSV curves of $Au_4Cu_2$ in 1 M KOH with and without glucose. **f** The corresponding potentials of GOR and OER at various current densities. **g** Chronoamperometric tests under a constant glucose concentration. **h** Chronoamperometric stability tests and the corresponding FEs of $Au_4Cu_2$ at 1.0 V vs. RHE for 40 cycles. All the potentials are presented without iR correction. Source data for this figure are provided as a Source data file.

Subsequently, the products of glucose electrooxidation were identified and quantified by high performance liquid chromatography (HPLC) with ultraviolet detector (Supplementary Fig. 12). $Au_4Cu_2$ alloy displays high yields and Faraday efficiency (FE) of PGA (>80%) over a wide potential range from 0.6 to 1.2 V vs. RHE (Fig. 2c), and the yield of PGA can reach up to 93.47% at 0.7 V vs. RHE. $Au_4Cu_2$ also exhibits higher PGA productivity compared with $Au_5Cu_1$ and $Au_3Cu_3$ (Supplementary Fig. 13). The yield of PGA gradually increases as the GOR proceeded (Fig. 2d and Supplementary Fig. 14), while the corresponding FE slightly decreases. Notably, $Au_4Cu_2$ achieves a high FE of 93.40%, PGA selectivity of 97.15%, along with the generation of small amounts of glucaric acid (2.33%), lactic acid (0.24%), and formic acid (0.10%) (Supplementary Figs. 15–16). As glucose undergone degradation in alkaline solutions[39], we investigated this process under the experimental conditions of GOR (Supplementary Fig. 17). The degradation of glucose is quantified at 3.20% at the same time as the experiment of GOR at 0.7 V vs. RHE. The acidic products from glucose degradation are limited, and the yields of gluconic acid, lactic acid and formic acid are only 0.10%, 0.05% and 0.05%, respectively. Thus, we can attribute the formation of PGA primarily to the electrocatalytic glucose oxidation. In addition, carbon mass balance during GOR process at 0.7 V vs. RHE is calculated to be 99.7%. In comparison to OER of $Au_4Cu_2$, the presence of glucose can decrease the demanded potential

by 1360 and 1280 mV at the current densities of 100 and 500 mA cm⁻², respectively (Fig. 2e), indicating that $Au_4Cu_2$ performs well for GOR. Figure 2f summarizes the current densities of GOR and OER on $Au_4Cu_2$ achieved at different potentials. The potential reductions between GOR and OER exceed 1.06 V from 50 to 500 mA cm⁻², implying that GOR is a desired substitute for OER at industrial water electrolysis.

The GOR durability was evaluated using a chronoamperometry method. In order to eliminate the influence of glucose consumption on the current density, a home-made reactor was designed to keep a constant glucose concentration for glucose electrooxidation by continuous supply of flowing electrolyte (Supplementary Fig. 18). The current density of Au is continuously decreasing over time at 0.7 V vs. RHE (Fig. 2g). Notably, the current density for Au catalyst decreases more rapidly as the reaction potential increasing to 1.0 V vs. RHE, suggesting a deactivation of Au at high potential. The quick deactivation at high potential is related to the surface oxide of Au-OH species to $AuO_x$ on Au catalyst[25,30]. By contrast, the current density for $Au_4Cu_2$ alloy remains stable at 1.0 V vs. RHE. It is reported that the deactivation of Au catalyst at 0.7 V vs. RHE was caused by the adsorbed linearly gluconic species rather than $AuO_x$[40]. $Au_4Cu_2$ alloy also exhibits a maintained performance at 0.7 V vs. RHE, demonstrating a high poisoning tolerance during glucose electrooxidation. The enhancement in durability may be credited with the strongly charge-polarized $Au^\delta$

$^{-}$-Cu$^{\delta+}$ sites, which are conducive to the selective adsorption of electronegative OH$^{-}$ on Cu$^{\delta+}$ sites, thereby inhibiting the formation of Au-OH species and the further oxidation. Following intermittent chronoamperometry tests, were carried out to assess the feasibility of practical application. The decreases of current densities in each cycle was caused by the consumption of glucose reactant. The current density can be recovered after refreshing the electrolyte each 3 h, with the FEs varies in the range of 95.44−99.99% for PGA production (Fig. 2h). Beyond that, no significant decreases in FEs can be detected during 40 cycles test, elucidating the robust stable of the Au$_4$Cu$_2$ alloy electrocatalyst.

Besides, a systematic study was carried out on the catalyst changes during GOR. The morphology and composition of Au$_4$Cu$_2$ alloy are maintained after a prolonged reaction (Supplementary Fig. 19 and Supplementary Table 1). XPS analyses reveal that valence states of Au in alloy remain essentially unchanged (Supplementary Fig. 20); meanwhile, zero-valent Cu is still observed on the alloy surface after GOR. And Au/Cu atomic ratio from XPS analysis increased from 1.14 to 1.30 after GOR. In order to accurately assess the changes in the metal composition during the reaction, we performed the time-of-flight secondary ion mass spectrometry (TOF-SIMS) for analysis of the outermost atomic layer of a surface. The surface Au/Cu ratio of alloy is 1.10, which changes slightly to 1.19 after GOR (Supplementary Fig. 21). A little alteration of surface Au/Cu ratio indicates the stability of AuCu surface. In addition, to evaluate the possibility that stability stems from the change of ECSA, we monitored the $C_{dl}$ of Au$_4$Cu$_2$ before and after GOR (Supplementary Fig. 22). The surface Au/Cu atomic ratio increased moderately from 6.16 to 6.64 mF cm$^{-2}$ after GOR. Combined with the stable surface composition, it is deduced that the catalytic stability of Au$_4$Cu$_2$ originates primarily from the preserved intrinsic activity of the AuCu surface.

## Insight into the electrochemical reaction mechanism

Electrochemical impedance spectroscopy (EIS) with different potentials was performed to investigate the kinetic process of GOR (Supplementary Fig. 23). The semicircular of Au$_4$Cu$_2$ becomes smaller as the potential increases, suggesting a decrease in impedance and faster reaction kinetics of GOR[41]. Moreover, Au$_4$Cu$_2$ shows the smaller semicircles in the Nyquist plots than that of Au, indicating the favorable electron transport rate of Au$_4$Cu$_2$. To gain in-depth understanding of excellent GOR performance and durability of Au$_4$Cu$_2$ alloy, in-depth electrochemical characterizations were conducted. The adsorption behavior of glucose was investigated by open-circuit potential (OCP), which could reflect the change of glucose absorbates on the inner Helmholtz layer[42,43]. A rapid decline occurs in OCP (1.17 V) of Au$_4$Cu$_2$ alloy after injecting 0.2 M glucose (Fig. 3a), surpassing those of Au (0.29 V) and Cu (0.14 V), which indicates a stronger affinity toward glucose over Au$_4$Cu$_2$ alloy. In situ Fourier transform infrared spectroscopy (FTIR) was employed to get insights into the mechanism of the glucose electrooxidation over Au$_4$Cu$_2$. The potential-dependent FTIR from 0.1 to 1.2 V vs. RHE provides critical information of the glucose oxidation pathways (Fig. 3b, c). Two peaks at 2924 and 2860 cm$^{-1}$ corresponding to C-H stretching vibration emerge on Au$_4$Cu$_2$ alloy, associated with glucose adsorption. The peak intensities of C-H stretching over Au$_4$Cu$_2$ are more evident than those for Au (Supplementary Fig. 24), indicating favorable adsorption of glucose for Au$_4$Cu$_2$ relative to Au, due to the charge-polarized Au$^{\delta-}$ sites. The slight attenuation of the glucose adsorption peaks indicates rapid replenishment of consumed glucose (Supplementary Fig. 25), ensuring a stable GOR process. Importantly, the distinct peaks appear on Au$_4$Cu$_2$ at 0.4 V vs. RHE from 1200 to 1800 cm$^{-1}$, which also increase at an enhanced potential. By contrast, the weaker peaks are detected for Au at 0.5 V vs. RHE. And no IR signal is received on pure Cu electrode at the same potentials (Supplementary Fig. 26). A peak at 1666 cm$^{-1}$ assigned to vibration band of H$_2$O appears before glucose oxidation. As the

reaction progresses, the bands at 1577 cm$^{-1}$ are attributed to the asymmetric stretching of COO$^{-}$[24,44], while the bands at 1410 and 1326 cm$^{-1}$ corresponds to symmetric stretching of COO$^{-}$[45,46]. The different vibration modes of COO$^{-}$ suggest the oxidation of glucose to gluconate. The vibration bands at 1755 cm$^{-1}$ belonging to C=O stretching of carbonyl species are detected on Au$_4$Cu$_2$[47,48], which is assigned to carbonyl intermediates (*CO-R) for glucose oxidation. *CO-R intermediates can be directly converted into gluconic acid by Au$_4$Cu$_2$ alloy, preventing the C-C bond cleavage of glucose and the generation of byproducts.

The adsorption of OH$^{-}$ also plays a significant role in glucose electrooxidation. The adsorption activity of OH$^{-}$ was investigated by the in situ FTIR spectra in 1 M KOH (Fig. 3d, e). A broad band around 3520 cm$^{-1}$ is assigned to *OH on Au$_4$Cu$_2$, which is detected at 0.3 V vs. RHE. By contrast, the O-H vibration peak of Au emerges at higher potential of 0.4 V vs. RHE. In addition, Au$_4$Cu$_2$ exhibits higher OH vibration peak area than Au at the same potential (Fig. 3f), suggesting that more OH species are formed and accumulated on the surface of Au$_4$Cu$_2$ alloy. To further clarify that the change of OH vibrational region primary stems from the enhanced OH$^{-}$ adsorption rather than surface restructuring. We have performed Raman spectra of the Au$_4$Cu$_2$ alloy in 1 M KOH (Supplementary Fig. 27). Two bands at 683 and 532 cm$^{-1}$ also appear at 0.3 V vs. RHE, which belong to the bending mode of Cu-OH and the top site OH stretching mode[49], respectively. The above results indicate that alloy with charge polarization sites contributes to the adsorption and activation of OH$^{-}$. The OH groups on Au$_4$Cu$_2$ are further supported by the D-H isotope labeling experiment (Supplementary Fig. 28). The IR band of O-D of Au$_4$Cu$_2$ is observed at 2560 cm$^{-1}$ in D$_2$O and KOD solution at 0.8 V vs. RHE. Then the D-H exchange is conducted by replacing the solution with H$_2$O and KOH through a peristaltic pump. The O-D vibration peak remains after D-H exchange, indicating that the O-D vibration is assigned to *OD species of catalyst surface rather than D$_2$O. Such D-H exchange experiment confirms the formation of *OH species. Considering that *OH can be oxidized to hydroxyl radicals (•OH), the surface *OH is further detected indirectly by the concentration of •OH. Electron paramagnetic resonance (EPR) spectra of electrolyte were performed using 5,5-dimethyl-1-pyrroline N-oxide (DMPO) as the trapping agent of •OH. Four peaks assigned to DMPO-•OH radical are detected over Au$_4$Cu$_2$, Au, and Cu, suggesting the generation of •OH in alkaline electrolyte (Supplementary Fig. 29). And more •OH radicals produced by Au$_4$Cu$_2$ are confirmed by the stronger signal. More •OH that derived from the oxidation of *OH imply more surface *OH species on Au$_4$Cu$_2$. Noticeably, glucose triggers a significant decline of DMPO-•OH peak intensity, due to the consumption of *OH by glucose.

Charge polarization site of alloy was credited with the enhanced adsorption capacity for glucose and OH$^{-}$, we further investigated the electron transfer of alloy by Bader charge analysis (Fig. 3g and Supplementary Fig. 30). For Au$_4$Cu$_2$, net electron accumulation (indicated by blue atoms) occurs on Au atoms, whereas charge depletion is prominent for Cu atoms, resulting in the charge polarization in Au$_4$Cu$_2$. Furthermore, an external electric field of 0.1 V Å$^{-1}$ was applied to ELF for simulation of charge polarization effect of Au$_4$Cu$_2$ at high potential. As shown in Fig. 3h, electrons are more localized at Au atoms under the external electric field, implying that the strong charge polarization can persist at high potential. Hence, the favorable co-adsorption during GOR can be fulfilled with electronegative OH$^{-}$ bonding to electron-deficient Cu$^{\delta+}$ site and glucose bonding to electron-rich Au$^{\delta-}$ site.

OCP tests have confirmed the favorable adsorption of glucose over Au than Cu. The adsorption site of OH$^{-}$ was determined by time-dependent in situ FTIR spectra (Fig. 3i–k and Supplementary Fig. 31). The characteristic bands at 3560 cm$^{-1}$ correspond to *OH on the Au surface, while the bands of *OH of Cu appear at 3490 cm$^{-1}$. The IR peak position of *OH on Au$_4$Cu$_2$ manifests a distinct redshift from 3540 to

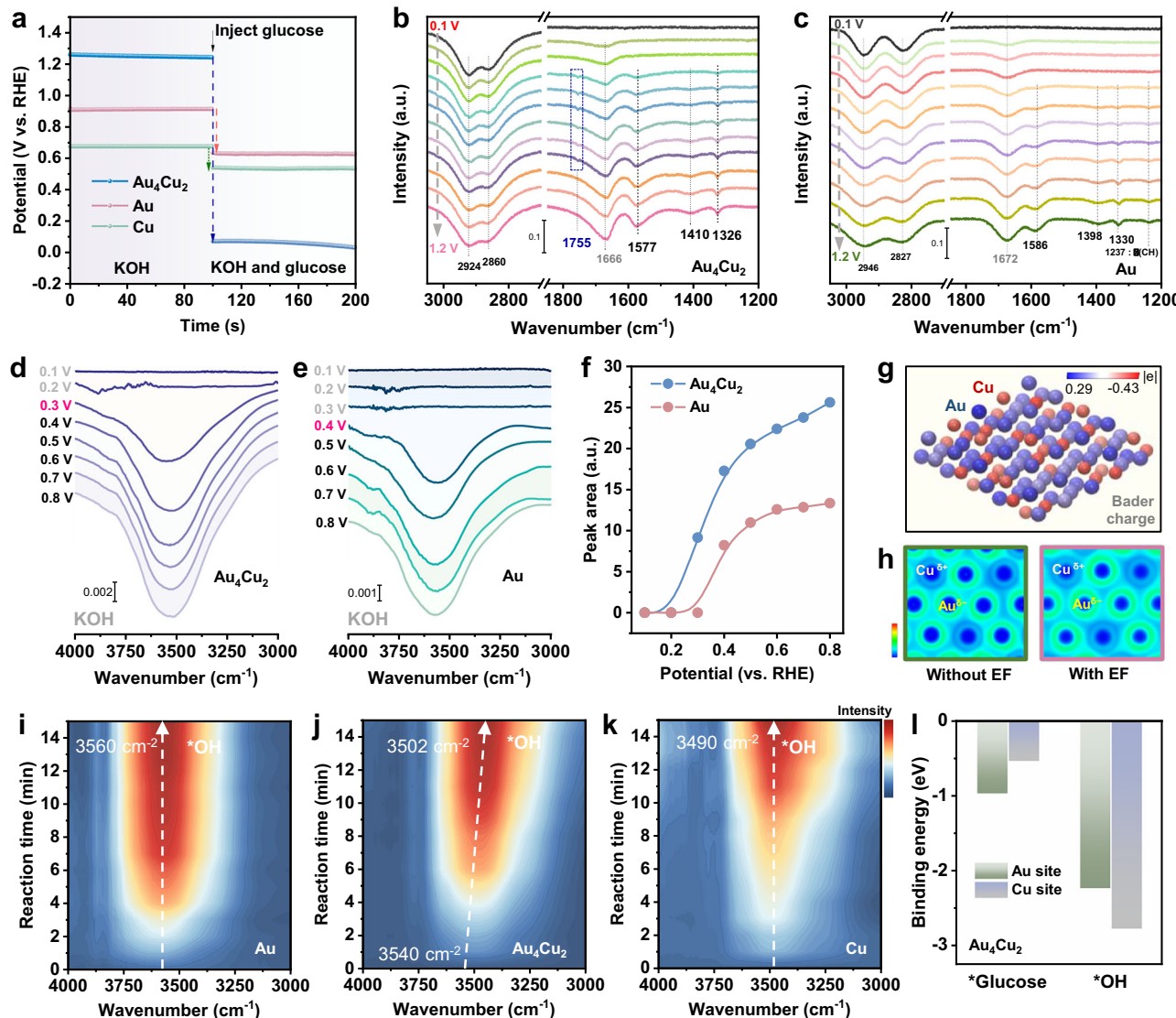

**Fig. 3 | GOR mechanism analysis by in situ and ex situ experiments. a** OCP tests in 1 M KOH before and after addition of glucose without iR correction. In situ electrochemical FTIR spectra of (**b**) Au₄Cu₂ and (**c**) Au for GOR in 1 M KOH with 0.2 M glucose. In situ FTIR spectra of (**d**) Au₄Cu₂ and (**e**) Au in 1 M KOH for adsorption of OH⁻. **f** Changes of peak area in of OH⁻ adsorption from in situ FTIR spectra. **g** Bader charge of Au₄Cu₂. **h** ELF of Au₄Cu₂ without and with external electric field (EF). Time-dependent in situ FTIR of (**i**) Au, (**j**) Au₄Cu₂, and (**k**) Cu in 1 M KOH with 0.2 M glucose. **l** Adsorption energy of *glucose and *OH on Au site and Cu site of Au₄Cu₂. Source data for this figure are provided as a Source data file.

$3502 \, cm^{-1}$ over time (Supplementary Fig. 32), which is inching to the *OH band of Cu. This phenomenon reveals that more *OH species are formed on $Cu^{\delta+}$ sites of Au₄Cu₂. The redshift of the hydroxyl peak in Au₄Cu₂ suggests a strong interaction between $Cu^{\delta+}$ site and hydroxyl, thereby leading to a weakening of the O-H bond strength[50]. Subsequently, the selective adsorptions of OH⁻ on Cu site and glucose on Au site were further verified by density functional theory (DFT) (Fig. 3l). The *OH bonded to the Cu atom of Au₄Cu₂ is significantly more negative (−2.77 eV) than that of the Au atom (−2.23 eV), suggesting a lower formation barrier of *OH on Cu atom. And glucose preferentially adsorbs on Au sites (−0.97 eV) rather than Cu sites (−0.53 eV). Based on the above results, Au₄Cu₂ alloy is capable of achieving exceptional glucose electrooxidation activity via the modulated co-adsorption of *OH and glucose.

DFT calculations were also conducted to theoretically rationalize the beneficial properties of Au₄Cu₂ for glucose oxidation. Typical structures and the adsorption configurations of molecules were constructed (Supplementary Figs. 33–40). Considering the pivotal role of co-adsorption of glucose substrate and OH species, we calculated the

adsorption energies for reactants on the surface of Au (111), Cu (111) and Au₄Cu₂ (111) (Fig. 4a). Au₄Cu₂ exhibits stronger adsorption energy for both *OH and glucose substrate than Au, which is conducive to the initial adsorption and activation of glucose. However, Cu shows a weak adsorption of glucose and strong adsorption of *OH, which is unfavourable for glucose oxidation reaction. Moreover, the binding strength of *CO-R key intermediates on Au₄Cu₂ is reduced compared to Au, which is due to the downshift of the d-band center of Au₄Cu₂. It is reported that lowering the d-band center can weaken the adsorption of carbonyl species and facilely release of CO from active sites[51,52]. DFT calculations offer valuable mechanistic insight at the atomic level. However, the computed results may deviate from the behavior of the actual catalyst, due to the challenges in modeling the exact material. Therefore, DFT calculations should be discussed alongside experimental results. Correspondingly, the projected density of states (PDOS) analysis reveals a downshift of the d-band center ($\varepsilon_d$) from −3.32 eV of Au to −3.55 eV of Au₄Cu₂ (Fig. 4b). Surface valence band photoemission measurements of AuCu alloy demonstrate that d-band center undergoes a downshift with increasing copper content (Fig. 4c).

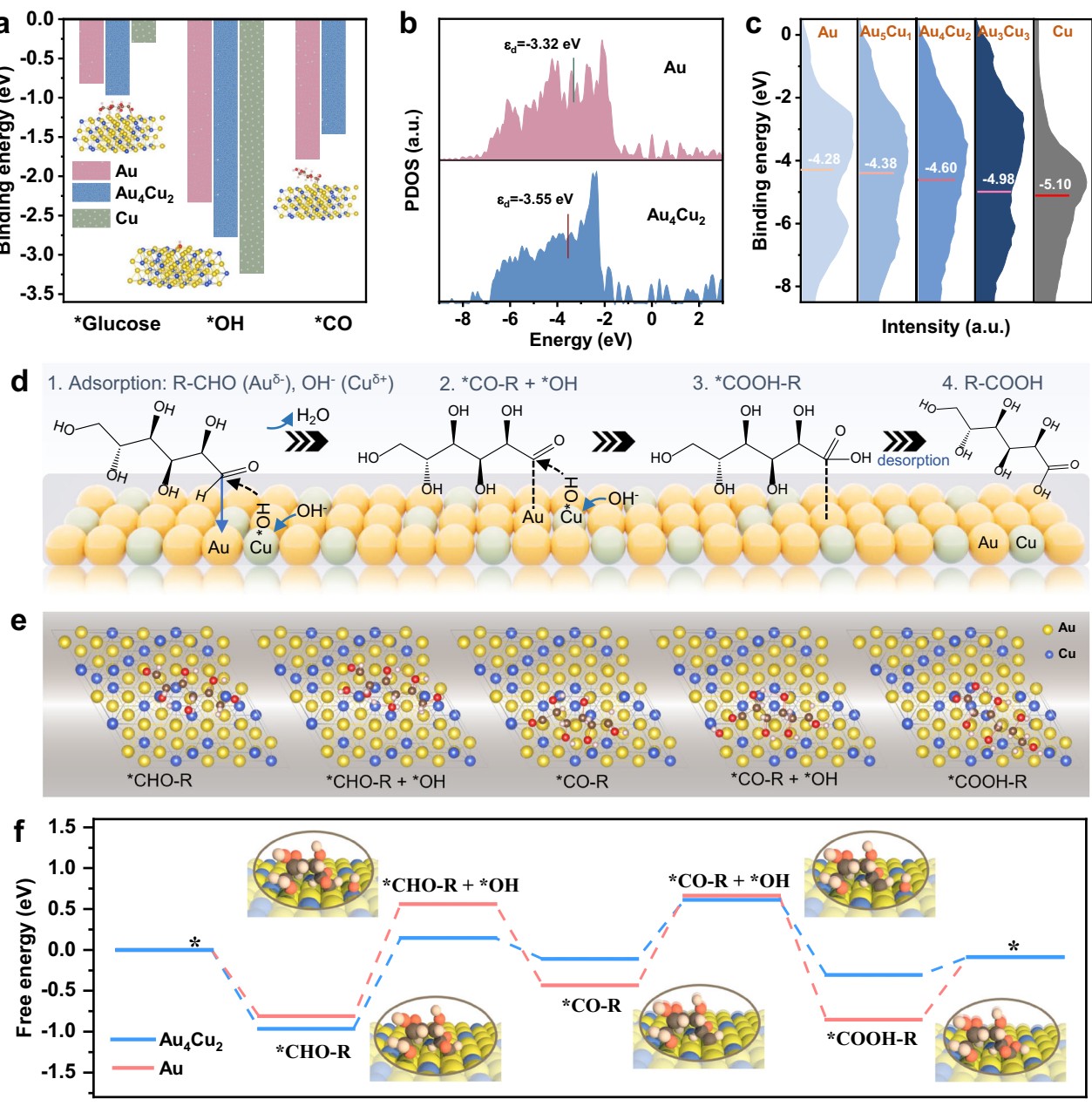

**Fig. 4 | DFT calculations. a** Adsorption energy of *glucose, *OH, and *CO-R on $Au_4Cu_2$, Au and Cu. **b** PDOS plots of $Au_4Cu_2$ and Au. **c** Surface valence band photoemission spectra. **d** Schematic diagrams of the proposed reaction pathway for glucose oxidation to PGA over $Au_4Cu_2$. **e** Optimized geometries of reaction intermediates configurations of molecules on $Au_4Cu_2$ (111). **f** Calculated free energy diagram of glucose oxidation on $Au_4Cu_2$ and Au. Source data for this figure are provided as a Source data file.

The alteration in the d-band center is related to the strength of intermediate adsorption[53]. The optimal d-band center position of $Au_4Cu_2$ can maximize GOR activity by regulating the desorption of intermediates. As a consequence, $Au_4Cu_2$ alloy with charge polarization sites can optimize the adsorption and desorption of reactants and intermediates during GOR.

Based on the aforementioned evidence, we can rationally deduce a plausible glucose electrooxidation reaction pathway on $Au_4Cu_2$, as shown in Fig. 4d. Firstly, the glucose undergoes spontaneous adsorption onto the catalyst surface, primarily via a carbonyl carbon rather than carbonyl oxygen atom. The positively charged carbonyl carbon of the aldehyde group tends to be attack by the electron-rich $Au^{\delta-}$ site (Supplementary Fig. 41). The generated *OH species can serve as active species to attack *glucose *via* direct oxidation of hydrogen of aldehyde group, leading to a formation of *CO-R as main intermediate. Subsequently, *CO-R is oxidized to *COOH-R by *OH, followed by its desorption and yield the final PGA product. Hence, the charge polarization of $Au_4Cu_2$ strategy facilitates adsorption of $Au^{\delta-}$-glucose and $Cu^{\delta+}$-OH, preventing the oxidation of Au-OH to $AuO_x$, thereby ensuring stable glucose electrooxidation. By contrast, the inevitably generated Au-OH species in Au catalyst surface are oxidized to $AuO_x$ at high potential, resulting in the Au deactivation (Supplementary Fig. 42). We further investigated the free energy diagrams of above reaction pathway on $Au_4Cu_2$ (111) and Au (111) (Fig. 4e, f). The oxidation reactions of intermediates and *OH proceed spontaneously for both $Au_4Cu_2$ and Au. The first formation of *OH active species is a rate-determining step

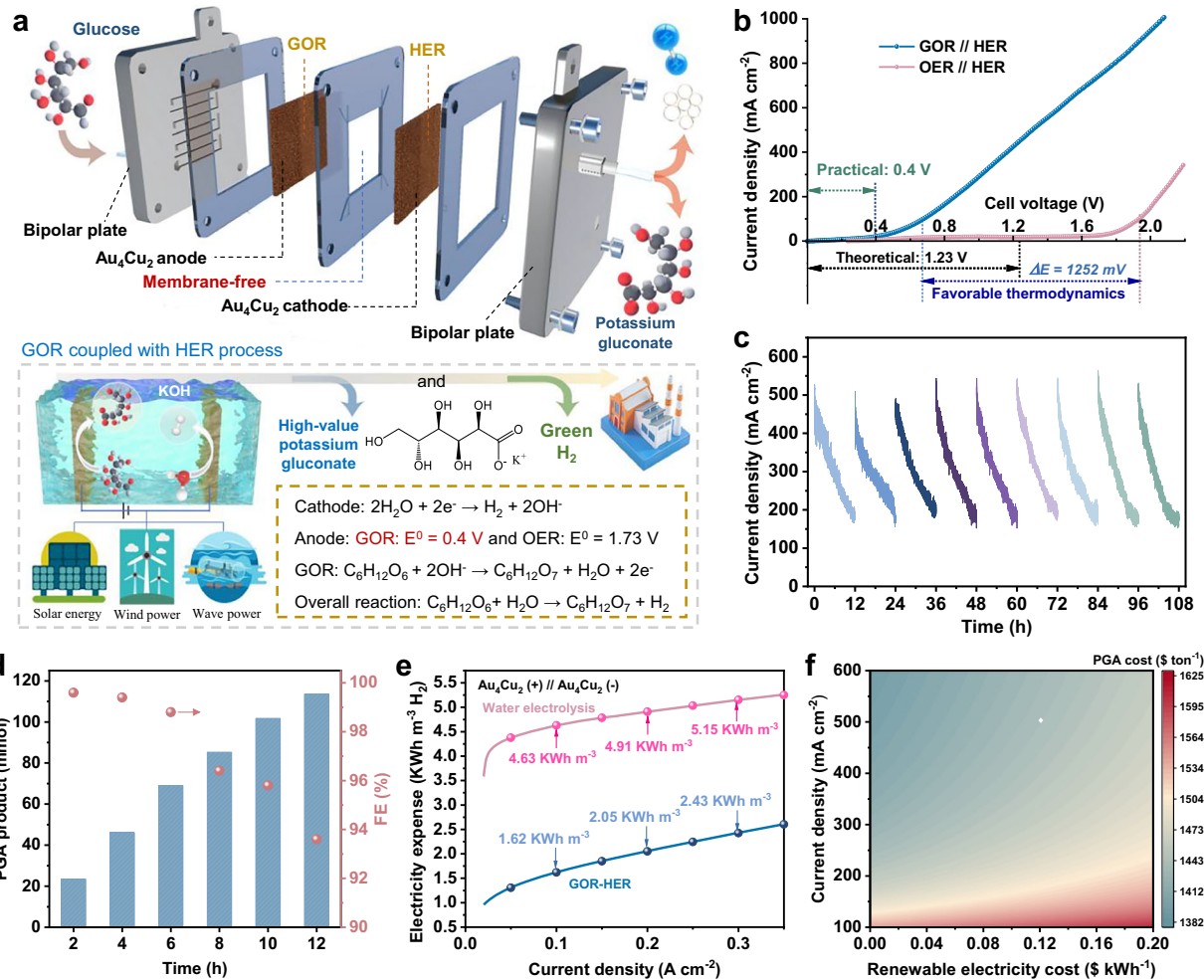

**Fig. 5 | Performance of two-electrode system. a** Schematic diagram of the membrane-free flow electrolyzer with $Au_4Cu_2$ alloy as anode and cathode, and direct GOR coupled with HER using renewable and intermittent electricity. **b** LSV curves of the membrane-free electrolyzer at a scan rate of $10\ mV\ s^{-1}$. **c** Stability evaluation at 1.2 V. **d** Products and FEs of membrane-free flow cell system at 1.2 V.

**e** Calculated electricity consumption for $H_2$ production. **f** TEA results of PGA cost against the current density and renewable electricity cost. Cell voltages are presented without iR correction. Source data for this figure are provided as a Source data file.

(RDS), facilitated by a reduced energy barrier on $Au_4Cu_2$ (1.11 eV), which is conducive to the subsequent deprotonation for glucose oxidation.

## Electrochemical property of membrane-free flow electrolyzer for GOR coupled with HER

Direct electrooxidation of glucose to PGA in tandem with $H_2$ production is a clean and sustainable alternative to fossil fuels, which can be driven by the renewable energy (Fig. 5a). Considering that high FE of GOR can inhibit oxygen generation, we assemble a membrane-free flow electrolyzer under two-electrode configurations consisting of $Au_4Cu_2$ alloy as anode and cathode with the working area of $1\ cm^2$ (Supplementary Fig. 43). LSV curves show that glucose electrooxidation occurs from 0.4 V and achieves the current density of $100\ mA\ cm^{-2}$ at 0.68 V, which is significantly advanced compared to water oxidation (1.94 V) (Fig. 5b). Moreover, the membrane-free electrolyzer outputted the current densities of 0.5 and $1\ A\ cm^{-2}$ at 1.31 V and 2.07 V for GOR, respectively. Such membrane-free coupled system demonstrates specific advantages in voltage input, compared with the reported organic electrooxidation coupled $H_2$ production systems (Supplementary Fig. 44 and Supplementary Table 3). The stability of electrolyzer was evaluated by intermittent potential tests. The membrane-free electrolyzer demonstrates reproducibility during durability testing,

whose current density can be recovered at 1.2 V over 108 h (Fig. 5c). Furthermore, membrane-free electrolyzer achieves PGA productivity of 113.5 mmol ($9.46\ mmol\ cm^{-2}\ h^{-1}$) with FE of 93.60% in a cycle (Fig. 5d). The above results indicate that membrane-free flow electrolyzer employing $Au_4Cu_2$ alloy electrode shows the potential of operation under industrially-relevant conditions.

The membrane-free flow electrolyzer of GOR and HER coupled system also has a cost-efficiency merits with lower electricity consumption of $1.62\ kWh\ m^{-3}\ H_2$ ($100\ mA\ cm^{-2}$), which is 65.0% lower than that of the water electrolysis system ($4.63\ kWh\ m^{-3}\ H_2$) (Fig. 5e). Noteworthily, compared to membrane-equipped electrolyzer, the membrane-free electrolyzer exhibits a higher current density under the same applied voltage (Supplementary Fig. 45). EIS analysis suggests that the membrane-free system shows a smaller solution resistance of about 1.18 Ohm, in contrast to the membrane-equipped system (1.83 Ohm) (Supplementary Fig. 46). Therefore, membrane-free electrolyzer not only reduces the capital expenditure, but also improves the electrocatalytic efficiency.

Techno-economic analysis (TEA) was conducted to further evaluate the economic feasibility of the system. The associated costs of industrial scale electrolysis and subsequent chemical processes are considered (Supplementary Table 4), including feedstock consumption, electricity cost, electrolyzer cost, installation and maintenance

cost, separation capital cost, etc. PGA production cost is linked with the operating current density of the electrolyzer (Fig. 5f). At 500 mA cm$^{-2}$, the production cost is calculated to be 1433\$ ton$^{-1}$ when the electricity cost is 0.12\$ kWh$^{-1}$. From a competitiveness perspective, renewable electricity price demonstrates limited effects on PGA production cost, which increased slightly from 1396\$ ton$^{-1}$ at 0.2\$ kWh$^{-1}$ to 1444\$ ton$^{-1}$ at 0.15\$ kWh$^{-1}$. Additionally, the net income from 1 ton of PGA production is calculated to be 712 \$ (Supplementary Fig. 47), with operating current density of 500 mA cm$^{-2}$ and electricity price of 0.12\$ kWh$^{-1}$, further highlighting the competitiveness and economic potential of the membrane-free GOR coupled with HER system.

## Discussion

In summary, a charge polarization strategy was employed to fabricate AuCu alloy with Au$^{\delta-}$-Cu$^{\delta+}$ sites as efficient catalyst for electrocatalytic glucose oxidation. The charge polarization is conducive to the co-adsorption with electronegative OH$^-$ bonding to electron-deficient Cu$^{\delta+}$ site and glucose bonding to electron-rich Au$^{\delta-}$ site, accelerating the formation of oxidative *OH and the dehydrogenation of *glucose to *CO-R intermediates. The incorporation of Cu also downshifts the d-band center of alloy, weakening the intermediate adsorption. Consequently, Au$_4$Cu$_2$ alloy achieves selective and stable glucose oxidation, with a low overpotential at 10 mA cm$^{-2}$ (0.14 V vs. RHE), high yield (93.47%) and selectivity (97.15%) of potassium gluconate. Finally, a membrane-free flow electrolyzer with Au$_4$Cu$_2$ as bifunctional electrode is designed to realize GOR and HER in parallel. And the integrated electrolyzer delivers a low cell voltage of 0.68 V at current density of 100 mA cm$^{-2}$, high productivity of potassium gluconate (9.46 mmol cm$^{-2}$ h$^{-1}$), as well as stable performance over 100 h. Our work delivers a technical direction for the development of cost-effective biomass upgrading as well as H$_2$ production in industrial applications.

## Methods

### Chemicals

All the chemicals were purchased from Aladdin and used without further purification, including copper sulfate pentahydrate (CuSO$_4$·5H$_2$O, 99.0%), Potassium gold(III) chloride (KAuCl$_4$, 98%), glucose (C$_6$H$_{12}$O$_6$, 99.0%), sulfuric acid (H$_2$SO$_4$), ethanol (99.8%), and potassium hydroxide (KOH, 95%). All solutions were prepared with ultrapure water (18.2 MΩ cm). Ni foam was purchased from Kunshan Guangjiayuan New Material Co., Ltd. The Fumasep FAB-PK-130 anion-exchange membrane was purchased from FuMa-Tech GmbH (Germany), which was soaked in 0.5 M NaCl aqueous solution for 48 h.

### Synthesis of Au$_x$Cu$_y$ alloy electrode

Au$_4$Cu$_2$ alloy was constructed on nickel foam (NF) through an electrodeposition method. In order to remove impurities and oxide layers from the NF surface, the NF was washed with hydrochloric acid (1.0 M), acetone, and ultrapure water by ultrasonication, respectively. Secondly, 30.2 mg of KAuCl$_4$ and 10 mg of CuSO$_4$·5H$_2$O were successively dispersed in 20 mL of 0.1 M sulfuric acid solution under stirring for 20 min (the molar ratio of Au to Cu is 4:2). Subsequently, the treated NF as the working electrode, Pt plate as the counter electrode, and Hg/Hg$_2$Cl$_2$ electrode as the reference electrode were immersed in the above mixture. The electrodeposition process was performed at −0.3 V (vs. SCE) for 10 min using CHI600A electrochemical workstation. The prepared Au$_4$Cu$_2$ alloy was cleaned with ultrapure water and then dried under vacuum at 60 °C for 1 h. The electrodes were named Au$_5$Cu$_1$ and Au$_3$Cu$_3$ were prepared by changing the molar ratio of initially added Au to Cu, while keeping the total addition amount of metal unchanged (the concentration of the metal is 6 mM). Au$_3$Cu$_3$. The preparation of Au and Cu electrodes followed the procedure described above without addition of CuSO$_4$·5H$_2$O and KAuCl$_4$, respectively.

## Materials characterizations

The Quattro S (Thermo Fisher Scientific, Germany) microscope was used to obtain scanning electron microscopy (SEM) images. Transmission electron microscopy (TEM) images were acquired on a Quanta 200 FEG operated at 200 kV (Philips FEI, USA). High-angle annular dark-field scanning transmission electron microscopy (HAADF-STEM) images and energy-dispersive X-ray spectroscopy (EDX) mapping images were conducted on Thermo Scientific Spectra 300. X-ray diffraction (XRD) spectra were recorded on a Bruker D8 Advance X-ray diffractometer with Cu Kα radiation (20 kV, $\lambda = 0.154056$ nm). The surface chemical composition was analyzed by X-ray photoelectron spectroscopy (XPS) measurements (ESCALAB 250, Thermo Fisher Scientific, Germany) using an Al Kα source (1486.6 eV) as a radiation source. And the d-band centre of the electrocatalysts was investigated by valence-band X-ray photoelectron spectroscopy (VB-XPS). The surface charge effect of the sample was corrected and normalized using the C 1s peak (284.8 eV) of the contaminated carbon as the internal standard. Electron paramagnetic resonance (EPR) spectroscopy was conducted on a Bruker EMXplus-6/1, and before the EPR test, the catalysts were activated by CV tests range from 0.2 V to 1.4 V vs. RHE for 20 cycles and then tested at 0.7 V vs. RHE for 5 min. Element content was detected by inductively coupled plasma-optical emission spectrometry (ICP-OES, Agilent 5110). Raman spectroscopy was recorded on a confocal Raman microscope (LabRAM Aramis, Horiba, Japan) with an excitation wavelength of 532 nm. The surface Au/Cu ratio of the catalyst was detected by time-of-flight secondary ion mass spectrometry (TOF-SIMS, PHI nanoTOF II Time-of-Flight SIMS, ULVAC-PHI. INC) with a 30 keV Bi$_3^{++}$ primary ion.

## Electrocatalytic measurements

The electrochemical measurements were conducted on a CHI 660E electrochemical workstation (Chenhua, Shanghai) in a three-electrode configuration at 25 °C. Ag/AgCl and Pt plate were used as the reference electrode and counter electrode, respectively. The acquired potentials were converted to values relative to the reversible hydrogen electrode (RHE). Linear sweep voltammetry (LSV) was conducted at a scan rate of 2 mV s$^{-1}$ under vigorous stirring. Electrochemical in situ electrochemical impedance spectroscopy (EIS) was conducted in a frequency range from 100 kHz to 0.1 Hz at different potentials, with an applied amplitude of 5 mV. Chronoamperometric stability tests of electrocatalyst were performed in 50 mL of alkaline solution (0.2 M glucose + 1 M KOH, pH = 13.8). All the electrochemical curves were used without IR compensation. The quasi-steady state LSV (QS-LSV) curve was carried out to accurately assess the GOR activity, and each point referred to the stabilized current density was recorded at a certain potential from the I-t test. The steady state Tafel slopes were recorded from equilibrated potentials under different current densities with a stabilization time of over 400 s. The CV curves in the non-Faradaic potential range were tested with different scan rates (50, 60, 70, 80, 90, and 100 mV s$^{-1}$) to calculate the effective ECSA of the catalyst. Using the following equation to calculate ECSA, where C$_s$ was the specific capacitance of a metallic surface (40 μF cm$^{-2}$), C$_{dl}$ was the double-layer capacitance:

$$\text{ECSA} = \frac{C_{dl}}{C_S} \tag{1}$$

The membrane-free flow cell was performed in a custom-made cell with flow alkaline solution (0.2 M glucose + 1 M KOH) as the electrolyte. The electrolyte solution was fixed at 1 L, and a peristaltic pump was utilized to close-circulate the solution at a flow rate of 100 mL min$^{-1}$. Au$_4$Cu$_2$ alloy with area of 1 × 1 cm$^2$ was used as the cathode and anode.

## Calibration method of reference electrode

The standard experimental calibration at 25 °C was conducted to maintain the accuracy of Ag/AgCl reference. Pt foil was both the working electrode and counter electrode, and 1 M KOH with 0.2 M glucose was selected as the electrolyte, which was saturated with high-purity hydrogen for 30 min. CV was carried out and centered at the open circuit potential with a scan range of ±0.2 V and a scan rate of 1 mV s$^{-1}$. The calibrated reference electrode potential was determined by taking the average of the two potentials corresponding to the zero point of current density.

## Product quantification

The glucose electrooxidation performances at different potentials were measured in 8 mL of 1 M KOH with 0.2 M glucose. The products were detected by a high-performance liquid chromatography (HPLC, LC5090Plus, Fuli Instruments) equipped with a UV detector at a wavelength of 210 nm. The detection temperature was 35 °C, the mobile phase was 5.0 mM $H_2SO_4$, and the flow rate was 0.6 mL min$^{-1}$. The chromatographic column used was Bio Rad Aminex HPX-87 (300 × 7.8 × 9 μm). Before the test, 0.1 mL of the liquid product was added to 0.9 mL of 55.6 mM $H_2SO_4$ solution for neutralization from 1 M KOH. Before the test, 0.1 mL of the liquid product was added to 0.9 mL of 55.6 mM $H_2SO_4$ solution for neutralization from 1 M KOH, which could also reduce the matrix effect. The measurements were only performed once. The yield of the PGA product was calculated by the following equations:

$$PGA\ yield(\%) = \frac{Moles\ of\ PGA\ in\ product}{Moles\ of\ glucose\ in\ feedstock\ input} \times 100\% \quad (2)$$

Faraday efficiency (FE) was calculated by the following equation, where $Q$ is the total charge that has passed through the electrode, $n$ (2) is the number of electron transfer for gluconic acid product from glucose, F is the Faraday constant (96485 C mol$^{-1}$):

$$FE(\%) = \frac{n \times F \times Moles\ of\ product}{Q} \times 100\% \quad (3)$$

## In situ electrochemical FTIR reflection spectroscopy

In-situ Fourier transform infrared spectroscopy (FTIR) spectroscopy was performed on a Perkin Elmer Spectrum 3. A custom-made thin-layer infrared cell with a standard three electrodes system was used for in situ FTIR test. The as-synthesized electrocatalysts were used as the working electrode, and Ag/AgCl and platinum wire were employed as the reference electrode and the counter electrode, respectively. The working electrode was directly pressed onto the CaF$_2$ optical window, and the infrared light beam passed through the CaF$_2$ window and the electrolyte, then reflected the infrared light on the electrode surface. In situ FTIR tests were collected from 0.1 to 1.2 V (vs. RHE) with an interval of 0.1 V. Before each test, a background spectrum of the electrocatalyst electrode was obtained at open circuit potential.

## Computational methods

The first principle calculations are performed by the Vienna Ab initio Simulation Package (VASP) with the projector augmented wave (PAW) method[54,55]. The exchange-functional is treated using the Perdew-Burke-Ernzerhof (PBE) functional[54]. in combination with the DFT-D3 correction[56]. The calculations were performed in a spin-polarized manner. The cut-off energy of the plane-wave basis is set at 400 eV. The Brillouin zone integration is accomplished with 3*3*1 Monkhorts-Pack K point sampling to optimize geometry and lattice size[57]. The self-consistent calculations apply a convergence energy threshold of 10$^{-5}$ eV. The equilibrium geometries and lattice constancies are optimized with maximum stress on each atom within 0.05 eV/Å. The Bader charge analysis was carried out using the method described by Henkelman et al.[58] The atomic coordinates of the optimized computational models were provided in CIF format (Supplementary Data 1).

## Data availability

Data that support the findings of this study can be found in the Article and its Supplementary Information. Source data are provided with this paper.

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

## Acknowledgements

Y.L. discloses support for the research of this work from National Natural Science Foundation of China (22309030), Guangdong Basic and

Applied Basic Research Foundation (2023A1515012589; 2026A1515012886). K.Z. discloses support for publication of this work from the Key Project of the Guangdong Provincial Department of Education (2024ZDZX3056). F.P. discloses support for publication of this work from Science and Technology Planning Project of Guangzhou City (2023A03J0026). X.T., C.H. and B.D. declares no relevant funding.

## Author contributions

F.P. led the project. Y.L. designed and performed most of the experiments and analyzed most of the data including material synthesis, characterization, and electrochemical tests. X.T. took the SEM and TEM images. C.H. and K.Z. performed the in situ electrochemical FTIR reflection spectroscopy. B.D. performed the theoretical calculations. Y.L., B.D., and F.P. wrote the paper. All authors discussed the results and commented on the manuscript.

## Competing interests

The authors declare no competing interests.
