## [Transparent Peer Review file · Nature Communications]

Regulating adsorption selectivity by charge-polarized Au δ^- -Cu δ^+ site for stable glucose electrooxidation

Corresponding Author: Professor Feng Peng

Version 0:

Reviewer comments:

Reviewer #1

(Remarks to the Author)

This study presents an electronegativity-driven charge redistribution strategy to design a charge-polarized Au δ^- -Cu δ^+ site on Au₄Cu₂ alloy, the resulting catalysis showed excellent electrochemical glucose oxidation. Although DFT adsorption energy calculation reveals the co-adsorption on charge-polarized Au δ^- -Cu δ^+ site of OH⁻ on electron-deficient Cu δ^+ sites and glucose on electron-rich Au δ^- sites, however, lacking relative experimental characterization to confirm this point. For direct electrochemical glucose oxidation for Au, Au-OH serves as the real active phase due to the strong adsorption capacity of hydroxyl; thus, it is needed to supplement relative experience to elucidate adsorption sites. According to the theory of mutual attraction of opposite charges, the authors proposed that negatively charged hydroxyl groups are easily adsorbed on positively charged Cu δ^+ sites. However, they ignored the fact that glucose is prone to deprotonation in an alkaline environment, resulting in the overall glucose molecule showing electronegativity. Therefore, glucose, like hydroxyl groups, tends to adsorb on the Cu δ^+ site rather than the Au δ^- site. Moreover, this manuscript lacks detailed characterizations of pure Cu, including in situ electrochemical IRAS spectra, OCP changes, CV curves of the adsorption and desorption of OH⁻, EPR, surface valence band photoemission spectra, adsorption energy calculation for both *OH and glucose substrate, etc. Therefore, at this stage, I do not recommend the publication of this manuscript in Nature Communications.

1. During the glucose oxidation process, as the voltage increases, glucose is consumed to generate potassium gluconate. Therefore, the peaks of glucose and *CO-R should show opposite downward and upward enhancement, respectively. The author needs to further check the IRAS spectra. Additionally, the infrared peaks of glucose are crucial information and should be placed in Figures 4e and 4f.

2. For the electrochemical glucose oxidation on noble metal, the active species is Au-OH. There is a lack of direct experimental evidence that hydroxyl is adsorbed on Cu rather than on Au. In the IRAS spectra of Figure S17, can two distinct peaks related to the hydroxyl groups in Au₄Cu₂ be attributed to Au-OH and Cu-OH, respectively?

3. To further determine that the active phase for electrochemical glucose oxidation is Cu-OH rather than Au-OH, EPR characterizations of pure Cu and Au with and without glucose are required to be provided.

4. The author mainly intends to illustrate that hydroxyl groups and glucose molecules are adsorbed at the Cu and Au sites, respectively. Thus, characterization of pure Cu is necessary, including in situ electrochemical IRAS spectra, OCP changes, CV curves of OH⁻ adsorption and desorption, EPR, surface valence band photoemission spectra, adsorption energy calculation for both *OH and glucose substrate, etc.

5. When passing through a small amount of charge, glucose can not completely convert into gluconic acid. However, there is no signal peak of glucose on the HPLC spectra, which puzzled me. Currently, it is very difficult to distinguish and quantify the mixture of glucose, gluconic acid, and glucaric acid. Therefore, the HPLC spectra of the standard substances of glucose, glucose acid, and glucose diacid are required, as well as the possible low-carbon intermediate, such as formic acid, etc.

6. There is no Bi element in the manuscript. However, in the relevant descriptions of Figures S11 and S7, the authors mentioned that electron transfer occurred from Bi to Au. This requires further checking.

Reviewer #2

(Remarks to the Author)

The manuscript presents a study on the catalytic performance of AuCu alloys for glucose electrooxidation, integrating

electrode characterization, mechanistic insights via IRAS, DFT calculations, and electrochemical analysis. The authors propose that Au δ^- —Cu δ^+ sites within the alloy serve as stable and efficient catalysts for the selective conversion of glucose to potassium gluconate at high current densities.

The topic is timely, and the results are promising. However, several critical issues must be addressed before the manuscript is suitable for publication.

It is well established that copper can dissolve in alkaline media at potentials above 0.3 V or 0.4 V vs RHE, which may compromise the long-term integrity of the catalyst, as indicated by the Pourbaix diagram of copper. This concern appears to be supported by the data in Figure S13: the Cu peak is notably weak after the glucose oxidation reaction (GOR), suggesting that most of the copper may have leached from the surface. This raises important questions regarding the stability of Cu under the reported reaction conditions.

Figure 3b: The cyclic voltammogram (CV) recorded in the absence of glucose is interpreted as showing OH $^-$ adsorption with a lower onset potential and increased intensity for Au $_4$ Cu $_2$ compared to Au. However, this signal could also be attributed to copper oxidation and dissolution into CuO $_2^{2-}$. The authors should consider alternative interpretations and provide additional data to support their assignment.

The manuscript would benefit from including electrochemical activity data for pure copper electrodes to benchmark the performance of the AuCu alloys. Determination of the electrochemical surface area (ECSA) is essential for comparing catalytic activity. However, this is complicated by potential copper dissolution, which may artificially increase the exposed Au surface and lead to misleading enhancements in activity. This issue should be addressed quantitatively or discussed in detail.

Additionally, glucose degradation in strongly alkaline solutions is a well-known phenomenon (e.g., DOI: 10.1139/v69-658). This should be discussed in the context of product selectivity and stability under the experimental conditions.

• Page 8, line 133: The sentence refers to "electron transfer from Bi to its surrounding Au atoms," yet Bi is not included among the reported materials. This appears to be an error and should be corrected or clarified.

• Tafel slopes: Are the reported slopes corrected for mass transport limitations and ohmic drop? If so, the method of correction should be clearly described. In addition, current densities in the Tafel plots should be normalized to the ECSA.

• The authors should incorporate recent references devoted to the glucose oxidation on Au electrodes, for example:

• 10.1007/BF03214967

• 10.1016/j.electacta.2022.140023

Reviewer #3

(Remarks to the Author)

The concept membrane-free flow electrolyzer of GOR and HER coupled system is not new, although the AuCu materials developed by Liu et al. leads to very impressive performances in terms of onset potential and achieved current densities. The high activity and stability of the catalyst is proposed to be due to "the strongly charge-polarized Au δ^- —Cu δ^+ sites". But, the experiments were done at 1.0 v vs RHE, and I am not sure that at so high potential, this effect is still present. In addition, authors propose a lot of explanations without any evidence for them. I think that the paper also lacks of convenient references for different aspects that are discussed by the authors considering the mechanisms of glucose oxidation on noble metals, ir spectroscopy, etc.

For example, it is said that the decline trend of current is caused by the consumption of glucose reactant. Can the authors demonstrate it? We have no information on the catalyst loading and electrode surface area, so it is difficult to check if it is due to glucose consumption of catalyst poisoning.

"Transition metal-based catalysts usually undergo structural reconstruction at high potentials during electrooxidation process, unavoidably resulting in C—C bond breakage and production of low-value C1 chemicals (e.g., formic acid)". Can the authors explain what is the relationship between structural reconstruction and C-C bond breaking?

Authors said "Previous investigations verified that noble metal catalysts (Au, Pt, and Pd) were widely used for biomass molecules electrooxidation to target product, owing to their moderate C-C bond-breaking capabilities." This is not true. It has clearly been shown recently by Faverge et al. that Pt and Pd were not selective and favored C-C bond breaking, but that Au was selective at low potentials. (See Faverge et al., ACS Catal. 13 (2023) 2657–2669).

Authors said "For alkaline GOR, the adsorbed glucose on the catalyst surface undergoes a dehydrogenation caused by *OH to form an intermediate, and the subsequent C=O activation followed by OH coupling". This mechanism is not true. It has again been shown recently by Neha et al. (Electrocatalysis 14 (2023) 121-130) and Faverge et al. (See Faverge et al., ACS Catal. 13 (2023) 2657–2669) that the oxidation of glucose could start in a potential range where OH does not adsorb on Au. Why chronoamperometry studies are performed at 1.0 V vs RHE and not at 0.7 V vs RHE? 1.0 V vs RHE is a very high potential, moreover, it is possible that the deactivation is less at high potential than at intermediate ones. The deactivation at intermediate potentials, 0.7-0.8 V vs RHE is not due to the formation of surface oxides, but according to Tominaga et al. (Electrochem. Commun. 9 (2007) 1892–1898) to adsorbed linearly gluconic species. Authors have to discuss this aspect.

Authors said "Fig. 3b describes the representative OH $^-$ adsorption and desorption curves. Au $_4$ Cu $_2$ shows an OH $^-$ oxidation adsorption peak with lower onset potential and increased intensity than that of Au." I think that this explanation is wrong. Indeed, can the authors explain how the desorption of OH $^-$ (reduction of the surface) could appear at higher potential than its oxidation (adsorption of OH $^-$)? The first oxidation peak is related with the reduction peak at ca. 0.3 V vs RHE, and is certainly due to the Cu/CuO redox couple!!!

The in situ infrared spectroscopy study does not give any information on the mechanism conversely to what explains the authors. They should refer to specialists in this domain. First, the fingerprint region of organics is between 800 and 2000 cm $^{-1}$. Here, nothing can be drawn from the IR study. It is very strange that no signal were recorded in the 1200 – 1600 cm $^{-1}$ region. "The vibration peaks at 1680 belonging to carbonyl intermediates (*CO-R) are detected on Au $_4$ Cu $_2$ "; This region is very close to the interfacial water vibration band (1640 cm $^{-1}$) and it is difficult to conclude. "characteristic vibration band of

carboxyl intermediates (COO⁻, 1535 cm⁻¹)” Carboxyl are generally observed around 1580 cm⁻¹, not 1535.

Minor

“Upon alloying Cu with Au, the diffraction peaks of Au₄Cu₂ shift obviously to a higher angle, which is ascribed to the lattice compression. Furthermore, the diffraction peaks shift to higher angle with an increasing amount of Cu, implying the formation of AuCu alloy.” Does it fit with the Vegard's law? And what is the level of alloying? Is it coherent with the other structural characterizations?

“The electron transfer from Bi to its surrounding Au atoms”. Bi ???

It is not clear what are the reactant? OH⁻ads or OH radical?

Reviewer #4

(Remarks to the Author)

The authors report an AuCu alloy that serves as an efficient electrocatalyst for the selective oxidation of glucose to potassium gluconate at high current density. The alloy enables co-adsorption of OH⁻ on electron-deficient Cu^{δ+} sites and glucose on electron-rich Au^{δ-} sites, promoting the formation of reactive *OH and key intermediates. Preferential OH adsorption on Cu^{δ+} suppresses Au–OH formation and subsequent oxidation to AuOx, mitigating deactivation. Au₄Cu₂ achieves 97.6% selectivity to potassium gluconate and reaches 500 mA cm⁻² at 0.88 V vs RHE. In a membrane-free flow electrolyzer, Au₄Cu₂ delivers stable operation with a potassium gluconate productivity of 9.46 mmol cm⁻² h⁻¹ and a Faradaic efficiency of 93.6%. However, this manuscript do not meet the requirement for publishing in Nat Comm before addressing the following issues.

1. Glucose is listed as C₆H₁₂O₂ (which should be C₆H₁₂O₆). This is a fundamental error in “Chemicals.”
2. Lattice spacings are reported in nm (e.g., 2.21 nm ... 2.37 nm), clearly meant to be Å;
3. The Faradaic efficiency formula is wrong as written: FE must be $FE = (nF \times \text{moles of product}) / Q$, but the equation shown omits the total charge Q and duplicates F . Also conflicting text says “ C is the total electron transfer”.
4. page 8, the differential discussion suddenly states “electron transfer from Bi to Au”, although the system is Au–Cu; this looks like copy-paste from another study.
5. Very high PGA selectivity (up to 97.6%) and FE (95–99.9%) are claimed, yet product analysis relies on UV-HPLC at 210 nm with 5 mM H₂SO₄ on an Aminex HPX-87 column, with no description of sample neutralization from 1 M KOH, matrix effects, lactone/hydrate equilibria, calibration quality, or a full carbon balance.
6. The mechanism repeatedly equates the surface *OH needed for GOR with solution •OH radicals detected by DMPO-EPR after electrolysis. DMPO distinguishes free radicals in solution, not the metal-bound *OH adsorbate that governs alkaline alcohol oxidation; the two are not interchangeable evidence.
7. All high currents are reported per geometric area of 3D Ni foam without ECSA or roughness factor; comparisons with planar controls (Au foil) are therefore not meaningful.
8. Authors state “no iR compensation” anywhere, yet emphasize unusually low voltages.
9. The LSVs at 2 mV s⁻¹ reaching hundreds of mA cm⁻² in concentrated alkali are strongly mass-transport limited, so they are not suitable for kinetic claims. should replace with galvanostatic polarization (steady current steps) and hydrodynamically controlled tests.
10. In this paper, the Au onset (~0.2 V vs RHE) is far below literature (~0.5 V vs RHE; ACS Catal. 2023, DOI: 10.1021/acscatal.2c05871); should reconcile by detailing reference calibration, pH/electrolyte, scan/hydrodynamics, iR compensation, and onset definition, and include side-by-side CVs.
11. O–H stretching near 3570 cm⁻¹ in water-alkali is ambiguous; D₂O substitution or isotopic labeling is needed to support assignment to activated OH rather than bulk water structure.

Version 1:

Reviewer comments:

Reviewer #1

(Remarks to the Author)

The authors replied well to my comments and questions. I think the manuscript can be accepted for publication.

Reviewer #2

(Remarks to the Author)

The manuscript has been improved following the revisions, but several of the responses provided are not fully convincing and leave important mechanistic questions unresolved. The additional data strengthen the case in part, yet ambiguities remain regarding the interpretation of surface stability and adsorption phenomena.

- Thank you for providing the ICP OES and stability data. While the bulk composition appears largely unchanged, it is important to note that ICP OES is not surface sensitive, and catalytic activity is governed primarily by the surface. The weak Cu signal observed in the XPS spectra before and also after GOR suggests possible surface depletion. In such a scenario, the apparent stability may not reflect intact AuCu alloying but could instead arise from (i) enhanced surface roughness due to partial Cu dissolution, which increases the electrochemically active surface area (ECSA), or (ii) Au₄Cu₂ alloy covered by an

Au rich overlayer, which may alter poisoning tolerance. To clarify the mechanism, additional surface sensitive or operando characterization would be highly valuable—for example, a more precise estimation of surface Cu content by comparing Au/Cu peak ratios in XPS, or complementary electrochemical surface area measurements. Such analyses would help distinguish whether the observed stable performance originates from genuine Au₄Cu₂ surface stability or from restructuring effects. This distinction is crucial, given that the DFT calculations presented assume the presence of both Cu and Au atoms in the top surface layer.

- The surface enhancement for Au₄Cu₂ is confirmed by the authors: “Double-layer capacitance (C_{dl}) of Au₄Cu₂ is measured to be 6.64 mF cm⁻², surpassing 4.94, 5.65 and 5.79 mF cm⁻² of Au, Au₅Cu₁, and Au₃Cu₃, respectively (Fig. S11).” In Fig. 2b, once normalized by ECSA, the activity of Au₄Cu₂ at low overpotential appears close to that of pure Au or other AuCu alloys. The difference in activity is observed above 0.5 V, where the current of Au₄Cu₂ continues to increase with potential.

- The in situ FTIR data are helpful, but it should be noted that Cu dissolution and surface restructuring could also modify the OH vibrational region. As a result, the observed differences between Au₄Cu₂ and Au may not solely reflect enhanced OH⁻ adsorption, but could partly arise from surface modification effects.

Reviewer #4

(Remarks to the Author)

I think the authors have mostly addressed the issues.

Version 2:

Reviewer comments:

Reviewer #2

(Remarks to the Author)

The authors have addressed most of the concerns raised.

RESPONSE TO REVIEWERS' COMMENTS

Reviewer #1:

This study presents an electronegativity-driven charge redistribution strategy to design a charge-polarized $\text{Au}^{\delta-}\text{-Cu}^{\delta+}$ site on Au_4Cu_2 alloy, the resulting catalysis showed excellent electrochemical glucose oxidation. Although DFT adsorption energy calculation reveals the co-adsorption on charge-polarized $\text{Au}^{\delta-}\text{-Cu}^{\delta+}$ site of OH^- on electron-deficient $\text{Cu}^{\delta+}$ sites and glucose on electron-rich $\text{Au}^{\delta-}$ sites, however, lacking relative experimental characterization to confirm this point. For direct electrochemical glucose oxidation for Au, Au-OH serves as the real active phase due to the strong adsorption capacity of hydroxyl; thus, it is needed to supplement relative experience to elucidate adsorption sites. According to the theory of mutual attraction of opposite charges, the authors proposed that negatively charged hydroxyl groups are easily adsorbed on positively charged $\text{Cu}^{\delta+}$ sites. However, they ignored the fact that glucose is prone to deprotonation in an alkaline environment, resulting in the overall glucose molecule showing electronegativity. Therefore, glucose, like hydroxyl groups, tends to adsorb on the $\text{Cu}^{\delta+}$ site rather than the $\text{Au}^{\delta-}$ site. Moreover, this manuscript lacks detailed characterizations of pure Cu, including in situ electrochemical IRAS spectra, OCP changes, CV curves of the adsorption and desorption of OH^- , EPR, surface valence band photoemission spectra, adsorption energy calculation for both *OH and glucose substrate, etc. Therefore, at this stage, I do not recommend the publication of this manuscript in Nature Communications.

Response: We are very grateful for your valuable comments and constructive suggestions on this research. In response to the issues you pointed out, such as the lack of experimental characterization, theoretical hypothesis flaws, and insufficient key data, we have carried out systematic revisions and supplementary studies.

(1) We have supplemented in situ FTIR spectra, which confirmed the selective adsorption model of " OH^- - $\text{Cu}^{\delta+}$ " and " $\text{glucose-Au}^{\delta-}$ " under alkaline conditions. Additionally, theoretical calculations were used to further compare the adsorption energies of hydroxyl groups, proving the rationality of the active phase mechanism.

(2) Regarding the issue you raised about the deprotonation of glucose under alkaline conditions leading to negative charge, we explain as follows:

1. Under alkaline conditions, glucose can lose a proton to first form an enediolate anion, which is an intermediate carrying a negative charge. However, this negatively charged intermediate is a strong nucleophile and will rapidly combine with protons, having an extremely short lifetime. Through the enediol anion intermediate, glucose can be isomerized to D-mannose or fructose. Hence, this electronegative intermediate cannot exist stably, which cannot have a strong competitive adsorption effect to OH^- on $\text{Cu}^{\delta+}$ site;

2. We have supplemented the electrostatic potential distributions of glucose and OH^- . First, the negative electrostatic potential of OH^- (-1.38 eV of O atom) can occupy the $\text{Cu}^{\delta+}$ sites rather than $\text{Au}^{\delta-}$ sites. While the carbonyl carbon of glucose carries a partial positive charge due to the highly electronegative oxygen atom, resulting in a decreased electron density around the carbon atom (1.39 eV). This positively charged carbonyl carbon of the aldehyde group tends to be attack by the electron-rich $\text{Au}^{\delta-}$ site for further deprotonation by *OH coupling. Therefore, OH^- preferentially occupies the $\text{Cu}^{\delta+}$ sites, while glucose adsorbs at the $\text{Au}^{\delta-}$ sites instead due to steric hindrance effects, further explaining the rationality of the selective adsorption;

3. Combined with DFT calculations, it was found that the electrostatic attraction energy between OH^- and $\text{Cu}^{\delta+}$ (-2.77 eV) is much greater than the interaction energy between glucose and $\text{Cu}^{\delta+}$ (-0.53 eV).

Scheme 1. Isomerization of glucose under alkaline conditions

Scheme 2. The electrostatic potential distributions of glucose

(3) Concerning the key characterization and comparative experiments of pure Cu, we have supplemented all the electrochemical and characterization tests you proposed for pure Cu. Through DFT calculations, we compared the adsorption energies of OH and glucose on pure Cu and Au₄Cu₂ alloy, clarifying the cooperative effect mechanism of Cu^{δ+} sites.

After the above-mentioned revisions, the experimental evidence chain and theoretical logic of this study have been significantly improved. We believe that the revised manuscript now meets the publication requirements of Nature Communications. We sincerely request that you review it again. The following is a detailed point-by-point response to the specific questions you raised.

Q1. During the glucose oxidation process, as the voltage increases, glucose is consumed to generate potassium gluconate. Therefore, the peaks of glucose and *CO-R should show opposite downward and upward enhancement, respectively. The author needs to further check the IRAS spectra. Additionally, the infrared peaks of glucose are crucial information and should be placed in Figures 4e and 4f.

Response: Thanks for the valuable comment. With the help of specialists and relevant literature, we reconducted the in situ FTIR study test, as shown in Fig. 3b and 3c. The FTIR results show different trends of IR peaks of glucose and intermediates. The peaks of intermediates enhanced as the potential increases, while the intensity of glucose peaks were slightly weakened. The peak of glucose would not completely disappear. The glucose consumed will be reabsorbed and replenished. The infrared peaks of glucose have been placed in Figures 3b and 3c. We have included corresponding content in the revised manuscript:

“Two peaks at 2924 and 2860 cm⁻¹ corresponding to C-H stretching vibration emerges

on Au_4Cu_2 alloy, associated with glucose adsorption. The peak intensities of C-H stretching over Au_4Cu_2 is more evident than that for Au (Fig. S22), indicating favorable adsorption of glucose for Au_4Cu_2 relative to Au, due to the charge-polarized $Au^{\delta-}$ sites. The slight attenuation of the glucose adsorption peaks indicates rapid replenishment of consumed glucose (Fig. S23), ensuring a stable GOR process. Importantly, the distinct peaks appear on Au_4Cu_2 at 0.4 V vs. RHE from 1200 to 1800 cm^{-1} , which also increases at an enhanced potential” (Page 14)

Fig. 3 In situ electrochemical FTIR spectra of (b) Au_4Cu_2 and (c) Au at different potentials for GOR in 1 M KOH with 0.2 M glucose

Fig. S23. FTIR spectra of Au_4Cu_2 at 0.1 V and 1.2 V vs. RHE during GOR.

Q2. For the electrochemical glucose oxidation on noble metal, the active species is Au-OH. There is a lack of direct experimental evidence that hydroxyl is adsorbed on Cu rather than on Au. In the IRAS spectra of Figure S17, can two distinct peaks related to the hydroxyl groups in Au_4Cu_2 be attributed to Au-OH and Cu-OH, respectively?

Response: Thanks for the valuable comment. In order to certify that hydroxyl is adsorbed on Cu rather than on Au. Time-dependent in situ FTIR spectra of Cu have been provided. The peaks at 3560 cm^{-1} correspond to Au-OH, while the peaks at 3490

cm^{-1} correspond to Cu-OH. The peak position of *OH on Au_4Cu_2 has a distinct redshift from 3540 to 3502 cm^{-1} , which is inching to the Cu-OH band (Fig. 3i-3k and Fig. S28 and S29). This phenomenon reveals that more *OH species are formed on $\text{Cu}^{\delta+}$ sites of Au_4Cu_2 . And the DFT has been used to determine the adsorptions of OH^- on Cu site and glucose on Au site (Fig. 3l). We have included corresponding content in the revised manuscript:

*“The adsorption site of OH^- was determined by time-dependent in situ FTIR spectra (Fig. 3i-3k and Fig. S28). The characteristic bands at 3560 cm^{-1} correspond to *OH on the Au surface, while the bands of *OH of Cu appear at 3490 cm^{-1} . The IR peak position of *OH on Au_4Cu_2 manifests a distinct redshift from 3540 to 3502 cm^{-1} over time (Fig. S29), which is inching to the *OH band of Cu. This phenomenon reveals that more *OH species are formed on $\text{Cu}^{\delta+}$ sites of Au_4Cu_2 . The redshift of the hydroxyl peak in Au_4Cu_2 suggests a strong interaction between $\text{Cu}^{\delta+}$ site and hydroxyl, thereby leading to a weakening of the O-H bond strength. Subsequently, the selective adsorptions of OH^- on Cu site and glucose on Au site were further verified by density functional theory (DFT) (Fig. 3l). The *OH bonded to the Cu atom of Au_4Cu_2 is significantly more negative (-2.77 eV) than that of the Au atom (-2.23 eV), suggesting a lower formation barrier of *OH on Cu atom. And glucose preferentially adsorbs on Au sites (-0.97 eV) rather than Cu sites (-0.53 eV). Based on the above results, Au_4Cu_2 alloy is capable of achieving exceptional glucose electrooxidation activity via the modulated co-adsorption of OH species and glucose.” (Page 16-17)*

Fig. 3. Time-dependent in situ FTIR of (i) Au, (j) Au_4Cu_2 and (k) Cu in 1 M KOH with 0.2 M glucose. (l) Adsorption energy of *glucose and *OH on Au site and Cu site of Au_4Cu_2 .

Fig. S28. In situ IRAS spectra of Au_4Cu_2 , Au and Cu at 1.0 V vs. RHE.

Fig. S29. Changes of peak position $^*\text{OH}$ from in situ IRAS spectra.

Q3. To further determine that the active phase for electrochemical glucose oxidation is Cu-OH rather than Au-OH, EPR characterizations of pure Cu and Au with and without glucose are required to be provided.

Response: Thanks for the valuable comment. EPR characterizations of pure Cu and Au with and without glucose have been provided in Fig. S26. Four peaks assigned to DMPO- $\cdot\text{OH}$ radical are detected over Au_4Cu_2 , Au and Cu, suggesting the generation of $\cdot\text{OH}$ in alkaline electrolyte. And more $\cdot\text{OH}$ radicals produced by Au_4Cu_2 are confirmed by the stronger signal. Glucose triggers a significant decline of DMPO- $\cdot\text{OH}$ peaks intensity of Au_4Cu_2 , Au and Cu. During the process of glucose oxidation, the reactant is the adsorbed hydroxyl species ($^*\text{OH}$) rather than OH radical. The decline of DMPO- $\cdot\text{OH}$ peaks may be due to the consumption of $^*\text{OH}$ by glucose. And EPR is mainly used as a means to detect free radicals. After careful consideration, we believe that using EPR cannot confirm the active phase for electrochemical glucose oxidation and can only be used as an auxiliary support. Therefore, we have performed the time-dependent

in situ FTIR spectra of Au₄Cu₂, Au and Cu to determine the adsorbed site of hydroxyl species, as shown in Fig. 3k-3m. We elaborated on this experiment in detail in Question 2 mentioned above. And the EPR characterizations also useful information: more •OH radicals produced by Au₄Cu₂ are confirmed by the stronger signal. More •OH that derived from the oxidation of *OH imply more surface *OH species on Au₄Cu₂ than Au and Cu.

Fig. S26. EPR spectra under different test conditions.

Q4. The author mainly intends to illustrate that hydroxyl groups and glucose molecules are adsorbed at the Cu and Au sites, respectively. Thus, characterization of pure Cu is necessary, including in situ electrochemical IRAS spectra, OCP changes, CV curves of OH⁻ adsorption and desorption, EPR, surface valence band photoemission spectra, adsorption energy calculation for both *OH and glucose substrate, etc.

Response: Thanks for the valuable comment and pointing out the lack of characterization of pure Cu. The characterizations of pure Cu including LSV curve (Fig. S9), in situ electrochemical IRAS spectra (Fig. S24), OCP changes (Fig. 3a), EPR (Fig. S26), surface valence band photoemission spectra (Fig. 4c), adsorption energy calculation for both *OH and glucose substrate (Fig. 4a), time-dependent in situ FTIR spectra (Fig. 3k)

LSV curve: Cu electrode shows a much poorer catalytic activity toward GOR compared with Au₄Cu₂ alloy electrode (Fig. S9), which has a high initial potential of 1.31 V.

In situ electrochemical IRAS spectra: And no IR signal is received on pure Cu electrode at the low potentials (Fig. S24).

OCP changes: A rapid decline occurs in OCP (1.17 V) of Au₄Cu₂ alloy after

injecting 0.2 M glucose (Fig. 3d), surpassing those of Au (0.29 V) and Cu (0.14 V), indicating superior affinity toward glucose over Au₄Cu₂ alloy.

Adsorption energy calculation for both *OH and glucose substrate: Au₄Cu₂ exhibits stronger adsorption energy for both *OH and glucose substrate than Au, which is conducive to the initial adsorption and activation of glucose. However, Cu shows a weak adsorption of glucose and strong adsorption of *OH, which is unfavourable for glucose oxidation reaction.

Surface valence band photoemission spectra: Surface valence band photoemission measurements of AuCu alloy demonstrate that d-band center undergoes a downshift with increasing copper content. The alteration in the d-band center is related to the strength of intermediate adsorption.

Fig. S9. LSV profiles of Au₄Cu₂ and Au electrocatalysts for glucose oxidation in 1.0 M KOH and 0.2 M glucose.

Fig. S24. In situ electrochemical FTIR spectra of Cu at different potentials for GOR in 1 M KOH with 0.2 M glucose (from 0.1 to 1.2 V vs. RHE).

Fig. 3a. OCP tests in 1 M KOH before and after addition of glucose.

Fig. 4a. Adsorption energy of *glucose, *OH, and *CO-R on Au₄Cu₂, Au and Cu. (b) PDOS plots of Au₄Cu₂ and Au.

Fig. 4c Surface valence band photoemission spectra.

Q5. When passing through a small amount of charge, glucose can not completely convert into gluconic acid. However, there is no signal peak of glucose on the HPLC spectra, which puzzled me. Currently, it is very difficult to distinguish and quantify the mixture of glucose, gluconic acid, and glucaric acid. Therefore, the HPLC spectra of the standard substances of glucose, glucose acid, and glucose diacid are required, as well as the possible low-carbon intermediate, such as formic acid, etc.

Response: Thanks for this insightful comment and pointing out the lack of necessary

validation data for HPLC of the standard substances. We have provided the standard curves of potassium gluconate, glucaric acid, and the possible low-carbon intermediate (glyceric acid, lactic acid and formic acid) in the revised manuscript (Fig. S12). The UV detector at a wavelength of 210 nm of liquid chromatography can only detect gluconic acid, but it cannot detect glucose, whose ultraviolet absorption is extremely weak, and there is no absorption at a wavelength of 210 nm. there is no signal peak of glucose on the HPLC when using UV detector. And we can quantify the products of gluconic acid and glucaric acid. When using the differential refractive index detector, we cannot distinguish and quantify the mixture, because the retention times of glucose and gluconic acid are very close. Hence, we use the UV detector to accurately quantify the products concentrations of glucose oxidation. Based on the results of the chromatography, we accurately obtained the yield and selectivity of the products. “The yield of PGA can reach up to 93.47% at 0.7 V vs. RHE. Au₄Cu₂ achieves a high FE of 93.40%, PGA selectivity of 97.15%, along with the generation of small amounts of glucaric acid (2.33%) and trace amounts of lactic acid (0.24%) and formic acid (0.10%) (Fig. S15 and S16).” (Page 11)

Fig. S12. The HPLC spectra of the standard substances and the corresponding calibration curves established with the external standards.

Fig. S16. The HPLC spectra of (a) potassium gluconate standard, and (b) potassium gluconate products during GOR.

Q6. There is no Bi element in the manuscript. However, in the relevant descriptions of Figures S11 and S7, the authors mentioned that electron transfer occurred from Bi to Au. This requires further checking.

Response: We sincerely thank the reviewer for pointing out this critical typo. The sentence should correctly describe as follows: "electron transfer from Cu to its surrounding Au atoms,". We have revised the sentence accurately in the revised manuscript.

Reviewer #2:

The manuscript presents a study on the catalytic performance of AuCu alloys for glucose electrooxidation, integrating electrode characterization, mechanistic insights via IRAS, DFT calculations, and electrochemical analysis. The authors propose that $\text{Au}^{\delta-}-\text{Cu}^{\delta+}$ sites within the alloy serve as stable and efficient catalysts for the selective conversion of glucose to potassium gluconate at high current densities.

The topic is timely, and the results are promising. However, several critical issues must be addressed before the manuscript is suitable for publication.

Response: It is a great honor to receive your positive remarks. Your constructive suggestions will be instrumental in refining the quality of this study, and we have carefully addressed all the points you raised.

Q1. It is well established that copper can dissolve in alkaline media at potentials above 0.3 V or 0.4 V vs RHE, which may compromise the long-term integrity of the catalyst, as indicated by the Pourbaix diagram of copper. This concern appears to be supported by the data in Figure S13: the Cu peak is notably weak after the glucose oxidation reaction (GOR), suggesting that most of the copper may have leached from the surface. This raises important questions regarding the stability of Cu under the reported reaction conditions.

Response: Thanks for the valuable comment. We performed ICP-OES analysis of Au_4Cu_2 before and after the stability tests of GOR (Table S1). The Cu content of Au_4Cu_2 before stability tests was 32.04%, which changed to 31.73% after stability tests. Hence, only traces of Cu leached from Au_4Cu_2 after the glucose oxidation reaction. As for the XPS spectra of Au_4Cu_2 before and after GOR, Au peak was also notably weak like Cu. It was probably because that the analyzed area of catalyst before and after GOR was different by ex situ XPS, which may have an influence on peak intensity. However, copper can be oxidated in alkaline media at potentials, and the leach of Cu can result in the deactivation of Au_4Cu_2 alloy during GOR. The pure Au surface exhibits catalytic instability for GOR. Therefore, we have performed a new stability study. The current density of Au is continuously decreasing over time (Fig. 2g). By contrast, the current

density of Au₄Cu₂ alloy remains stable, demonstrating a high poisoning tolerance during glucose electrooxidation. The stable GOR performance indicated that the leach of Cu was little. Hence, the stability of Cu under the reaction conditions was satisfactory.

Fig. 2g. Chronoamperometric tests under a constant glucose concentration.

Q2. Figure 3b: The cyclic voltammogram (CV) recorded in the absence of glucose is interpreted as showing OH⁻ adsorption with a lower onset potential and increased intensity for Au₄Cu₂ compared to Au. However, this signal could also be attributed to copper oxidation and dissolution into CuO₂²⁻. The authors should consider alternative interpretations and provide additional data to support their assignment.

Response: Thank you very much for pointing out this issue. You are right, and the first oxidation peak of CV can be assigned to the oxidation of Cu (*J. Am. Chem. Soc.* 2019, 141, 31, 12192-12196). Hence, it will be very difficult to obtain the correct result of the adsorption of OH⁻ by CV curves. Therefore, we have provided additional data. In situ FTIR spectra of Au₄Cu₂ and Au in 1 M KOH have been performed to investigate the adsorption of OH⁻ (Fig. 3d and 3e). The O-H vibration peak intensity of Au₄Cu₂ is higher than that of Au. And the *OH peak of Au₄Cu₂ can be detected at a lower voltage. Based on the new experimental data, we can conclude that Au₄Cu₂ alloy was credited with the enhanced adsorption capacity for OH⁻.

The corresponding discussion have been included in the revised manuscript:

*“The adsorption activity of OH⁻ was investigated by the in situ FTIR spectra in 1 M KOH (Fig. 3d and 3e). A broad band around 3520 cm⁻¹ is assigned to *OH on Au₄Cu₂,*

which is detected at 0.3 V vs. RHE. By contrast, the O-H vibration peak of Au emerges at higher potential of 0.4 V vs. RHE. In addition, Au_4Cu_2 exhibits higher O-H vibration peak area than Au at the same potential (Fig. 3f), suggesting that more OH species are formed and accumulated on the surface of Au_4Cu_2 alloy. The above results indicate that alloy with charge polarization sites contributes to the adsorption and activation of OH^- .” (Page 15)

Fig. 3. In situ FTIR spectra of (d) Au_4Cu_2 and (e) Au in 1 M KOH for adsorption of OH^- . (f) Changes of peak area in of OH^- adsorption from in situ FTIR spectra.

Q3. The manuscript would benefit from including electrochemical activity data for pure copper electrodes to benchmark the performance of the AuCu alloys. Determination of the electrochemical surface area (ECSA) is essential for comparing catalytic activity. However, this is complicated by potential copper dissolution, which may artificially increase the exposed Au surface and lead to misleading enhancements in activity. This issue should be addressed quantitatively or discussed in detail.

Response: Thanks for the valuable comment. LSV curve of pure Cu electrodes for glucose oxidation was exhibited in Fig. S9. “Cu electrode shows a much poorer catalytic activity toward GOR compared with Au_4Cu_2 alloy electrode (Fig. S9), which has a high initial potential of 1.31 V.” In addition, the characterizations of pure Cu including in situ electrochemical IRAS spectra (Fig. S24), OCP changes (Fig. 3a), adsorption energy calculation for both $*OH$ and glucose substrate (Fig. 4a) etc. have been provided.

Fig. S9. LSV profiles of Au_4Cu_2 and Au electrocatalysts for glucose oxidation in 1.0 M KOH and 0.2 M glucose.

In addition, we have estimated ECSA based on the double-layer capacitance (C_{dl}). We have included more discussion in the revised manuscript, as read: “*The electrochemical surface area (ECSA) was carried out through cyclic voltammetry (CV) with different scan rates (Fig. S10). Double-layer capacitance (C_{dl}) of Au_4Cu_2 is measured to be 6.64 mF cm^{-2} , surpassing 4.94 , 5.65 and 5.79 mF cm^{-2} of Au, Au_5Cu_1 , and Au_3Cu_3 , respectively (Fig. S11).*” (Page 10)

Fig. S10. CV curves of (a) Au, (b) Au_5Cu_1 , (c) Au_4Cu_2 and (d) Au_3Cu_3 at different scan rates from 50 to 100 mV s^{-1} in 1 M KOH and 0.2 M glucose.

Fig. S11. The extracted double-layer capacitances (C_{dl}) of different electrodes using a CV method.

Considering that the ECSA has a significant impact on catalytic activity, it leads to the misleading enhancements in activity. Quasi-steady-state linear sweep voltammetry (QS-LSV) was also carried out to accurately assess the GOR activity (Fig. 2b). In addition, the QS-LSV plots was normalized through the ECSA to assess the intrinsic activity of GOR. As shown, the Au_4Cu_2 demonstrated better electrocatalytic performance of GOR compared to Au, Au_5Cu_1 , and Au_3Cu_3 electrode. We have included corresponding content in the revised manuscript:

“To accurately assess the intrinsic activity of GOR, quasi-steady-state linear sweep voltammetry (QS-LSV) normalized through the ECSA is conducted based on the equilibrated current density under a given potential (Fig. 2b), where Au_4Cu_2 demonstrates the best catalytic activity of GOR.” (Page 10)

Fig. 2b ECSA-normalized QS-LSV plots toward GOR, inset is the steady state Tafel slopes.

Q4. Additionally, glucose degradation in strongly alkaline solutions is a well-known phenomenon (e.g., DOI: 10.1139/v69-658). This should be discussed in the context of product selectivity and stability under the experimental conditions.

Response: Thanks for the valuable comment. The potential degradation of glucose in strongly alkaline electrolytes is a crucial factor that must be considered and discussed to validate the reported high selectivity. Our electrocatalytic experiments are characterized by relatively short reaction times (1.47 h at 0.7 V vs. RHE) at 25 °C. The glucose degradation was investigated under the experimental conditions of GOR (Fig. S17). We have now addressed this issue in the revised manuscript, as detailed below.

“As glucose undergone degradation in alkaline solutions³⁹, we investigated this process under the experimental conditions of GOR (Fig. S17). The degradation of glucose is quantified at 3.20% at the same time as the experiment of GOR at 0.7 V vs. RHE. The acidic products from glucose degradation are limited, and the yields of gluconic acid, lactic acid and formic acid are only 0.10%, 0.05% and 0.05%, respectively. Thus, we can attribute the formation of PGA primarily to the electrocatalytic glucose oxidation. In addition, carbon mass balance during GOR process at 0.7 V vs. RHE is calculated to be 99.7%.” (Page 11)

Fig. S17. (a) HPLC spectra of glucose detected by differential refractive index detector, and (b) the corresponding calibration curves, (c) HPLC spectra of glucose degradation under the experimental conditions of GOR.

In order to eliminate the influences of glucose consumption and glucose degradation on the current density, we design a home-made reactor for glucose electrooxidation (Fig. S18). 1 M KOH with 0.2 M glucose solution was continuously fed into the reactor, which could keep a constant glucose concentration for glucose

electrooxidation. In this way, we can eliminate the influence of glucose consumption and glucose degradation on the current. The current density of Au is continuously decreasing over time (Fig. 2g). By contrast, the current density of Au₄Cu₂ alloy remains stable, demonstrating the durability of Au₄Cu₂ alloy. We have included the corresponding discussion in the revised manuscript.

Fig. S18. A home-made reactor for glucose electrooxidation under stirring, with continuous supply of flowing electrolyte.

Fig. 2g. Chronoamperometric tests under a constant glucose concentration.

Q5. Page 8, line 133: The sentence refers to "electron transfer from Bi to its surrounding Au atoms," yet Bi is not included among the reported materials. This appears to be an error and should be corrected or clarified.

Response: We sincerely thank the reviewer for pointing out this critical typo. The sentence should correctly describe as follows: "electron transfer from Cu to its surrounding Au atoms,". We have revised the sentence accurately in the revised manuscript.

Q6. Tafel slopes: Are the reported slopes corrected for mass transport limitations and ohmic drop? If so, the method of correction should be clearly described. In addition, current densities in the Tafel plots should be normalized to the ECSA.

Response: Thanks for the valuable comment. The reported slopes were not corrected for ohmic drop, and the electrochemical curves were used without IR compensation. To address the mass transport limitations, we have re-evaluated our Tafel plots and the steady state Tafel slopes were performed in the kinetically controlled region (inset of Fig. 2b). In addition, the current density was normalized through the ECSA.

The method of the steady state Tafel slopes have been clearly described in Methods. We have also included discussion of steady state Tafel slopes in the revised manuscript: “The steady state Tafel slopes were recorded from equilibrated potentials under different current densities with a stabilization time of over 400 s, and the current density was normalized through the ECSA.” (Page 25)

“A steady state Tafel slope analysis was further applied to assess the reaction kinetics during the GOR, and the current density was also normalized through the ECSA (inset of Fig. 2b). The Tafel plots of Au_4Cu_2 is $101.4 \text{ mV dec}^{-1}$, smaller than that of Au_5Cu_1 ($131.4 \text{ mV dec}^{-1}$), Au_3Cu_3 ($220.8 \text{ mV dec}^{-1}$), and Au ($169.1 \text{ mV dec}^{-1}$), revealing the faster GOR kinetics.”(Page 10-11)

Fig. 2b inset. the steady state Tafel slopes.

Q7. The authors should incorporate recent references devoted to the glucose oxidation on Au electrodes, for example:

- [10.1007/BF03214967](https://doi.org/10.1007/BF03214967)
- [10.1016/j.electacta.2022.140023](https://doi.org/10.1016/j.electacta.2022.140023)

Response: We appreciate the Reviewer 2 for providing these important literatures of glucose oxidation on Au electrodes. As recommended, we have carefully studied the two suggested references, and these references have been cited in the revised manuscript.

Reference 21: “[10.1007/BF03214967](https://doi.org/10.1007/BF03214967)”.

Reference 23: “[10.1016/j.electacta.2022.140023](https://doi.org/10.1016/j.electacta.2022.140023)”.

Reviewer #3:

The concept membrane-free flow electrolyzer of GOR and HER coupled system is not new, although the AuCu materials developed by Liu et al. leads to very impressive performances in terms of onset potential and achieved current densities. The high activity and stability of the catalyst is proposed to be due to “the strongly charge-polarized $\text{Au}^{\delta-}\text{-Cu}^{\delta+}$ sites”. But, the experiments were done at 1.0 v vs RHE, and I am not sure that at so high potential, this effect is still present. In addition, authors propose a lot of explanations without any evidence for them. I think that the paper also lacks of convenient references for different aspects that are discussed by the authors considering the mechanisms of glucose oxidation on noble metals, ir spectroscopy, etc.

Response: We are deeply grateful to the reviewer for taking the time to review our manuscript and providing constructive comments and suggestions. Your expert is highly constructive, and we appreciate the opportunity to provide further clarification and evidence. We took your comments as a guide to comprehensively improve the scientificity and persuasiveness of the manuscript by supplementing experimental data, improving theoretical analysis, and systematically sorting out references.

We have provided additional experiments for the strongly charge-polarized $\text{Au}^{\delta-}\text{-Cu}^{\delta+}$ sites at the high potential. An external electric field of 0.1 V \AA^{-1} was applied to ELF for simulation of charge polarization effect of Au_4Cu_2 at high potential. As shown in Fig. 3h, electrons are more localized at Au atoms under the external electric field, implying that the strong charge polarization can persist at high potential. In addition, the new chronoamperometric test at 1.0 V vs. RHE was conducted under a constant glucose concentration (Fig. 2g). The current density of Au decreases more rapidly as the reaction potential increasing to 1.0 V vs. RHE. By contrast, the current density of Au_4Cu_2 alloy remains stable at 1.0 V vs. RHE. The enhancement in durability may be credited with the strongly charge-polarized $\text{Au}^{\delta-}\text{-Cu}^{\delta+}$ sites, which is conducive to the selective adsorption of electronegative OH^- on $\text{Cu}^{\delta+}$ sites, thereby inhibiting the formation of Au-OH species and the further oxidation.

We have also provided a new IR spectroscopy for the mechanism study of glucose

oxidation on noble metals. We believe that by addressing your concerns, the overall quality of the manuscript have been significantly improved. Below, we have provided detailed point-by-point responses to all the issues raised.

Fig. 3. (h) ELF of Au_4Cu_2 without and with external electric field (EF).

Q1. For example, it is said that the decline trend of current is caused by the consumption of glucose reactant. Can the authors demonstrate it? We have no information on the catalyst loading and electrode surface area, so it is difficult to check if it is due to glucose consumption or catalyst poisoning.

Response: We thank the reviewer for the constructive suggestion. In order to determine whether the decrease in current is due to glucose consumption or catalyst poisoning, we design a home-made reactor for glucose electrooxidation, as shown in **Fig. S18**. 1 M KOH with 0.2 M glucose solution was continuously fed into the reactor through a peristaltic pump, which could keep a constant glucose concentration for glucose electrooxidation. In this way, we can eliminate the influence of glucose consumption on the current. The current density for Au catalyst is continuously decreasing over time (Fig. 2g). By contrast, the current density for Au_4Cu_2 alloy remains stable, demonstrating a high poisoning tolerance during glucose electrooxidation. We have included the corresponding discussion in the revised manuscript:

“In order to eliminate the influence of glucose consumption on the current density, a home-made reactor was designed to keep a constant glucose concentration for glucose electrooxidation by continuous supply of flowing electrolyte (Fig. S18). The current density of Au is continuously decreasing over time at 0.7 V vs. RHE (Fig. 2g). Notably, the current density for Au catalyst decreases more rapidly as the reaction potential increasing to 1.0 V vs. RHE, suggesting a deactivation of Au at high potential. The quick deactivation at high potential is related to the surface oxide of Au-OH species to AuO_x

on Au catalyst. By contrast, the current density for Au_4Cu_2 alloy remains stable at 1.0 V vs. RHE. It is reported that the deactivation of Au catalyst at 0.7 V vs. RHE was caused by the adsorbed linearly gluconic species rather than AuO_x . Au_4Cu_2 alloy also exhibits a maintained performance at 0.7 V vs. RHE, demonstrating a high poisoning tolerance during glucose electrooxidation.” (Page 12)

Fig. S18. A home-made reactor for glucose electrooxidation under stirring, with continuous supply of flowing electrolyte.

Fig. 2g. Chronoamperometric tests under a constant glucose concentration.

Q2. “Transition metal-based catalysts usually undergo structural reconstruction at high potentials during electrooxidation process, unavoidably resulting in C–C bond breakage and production of low-value C1 chemicals (e.g., formic acid)”. Can the authors explain what is the relationship between structural reconstruction and C–C bond breaking?

Response: We thank the reviewer for this question, which allows us to clarify the relationship between catalyst reconstruction and C–C bond cleavage. Generally

speaking, structural reconstruction alters the chemical nature, geometric configuration, and electronic properties of catalyst active sites, thereby modulating the adsorption modes of organic molecules, reaction pathways, and intermediate stability, ultimately promoting or inhibiting C-C bond cleavage. The intrinsic relationship between structural reconstruction of transition metal-based catalysts and C-C bond scission during electrooxidation processes is understood as follows:

(1) Structural reconstruction creates a highly electrophilic species

At high anodic potentials relevant to electrooxidation, transition metal (e.g., Ni, Co) surfaces undergo significant structural reconstruction. The surface metal atoms are oxidized to higher valence states (e.g., $\text{Ni}^{2+}/\text{Ni}^{3+}$, $\text{Co}^{2+}/\text{Co}^{3+}$), which is conducive to the generation of electrophilic oxygen species and the formation of $\text{M}^{3+}\text{-(OH)}_{\text{ads}}$. The $\text{M}^{3+}\text{-(OH)}_{\text{ads}}$ has electrophilicity and oxidizability. Their strong electrophilicity preferentially attacks $\alpha\text{-C}$ or $\beta\text{-C}$ sites in organic molecules (e.g., C1-C2 bond in glucose), inducing C-C bond cleavage. Hence, the structural reconstruction for the formation of $\text{M}^{3+}\text{-(OH)}_{\text{ads}}$ can result in the C-C bond breaking. For Ni-based catalysts, Wang et al. summarized this process: (1) the electrochemical structural reconstruction for generation of $\text{Ni}^{3+}\text{-(OH)}_{\text{ads}}$, (2) hydrogen atom transfer reactions ($\text{Ni}^{3+}\text{OOH} + \text{polyhydric alcohols} \rightarrow \text{Ni}^{2+}\text{(OH)}_2 + \text{Oxidation products}$), (3) $\text{Ni}^{3+}\text{-(OH)}_{\text{ads}}$ -induced C-C bond cleavage. (*Natl. Sci. Rev.* 2023, 10, nwad099). In our previous work, in situ infrared and Raman characterizations revealed that Ru-doped Ru-CoP₂ catalysts undergo dynamic reconstruction from Ru-CoP₂ to Ru-CoOOH during glycerol electrooxidation. This process reduces the C-C bond activation energy barrier, accelerating C-C cleavage in glucose and enabling superior formic acid production performance (*J. Energy Chem.*, 2025, 100, 317-326).

(2) Metal-metal bond modulation in alloy reconstruction regulates intermediate stability

Structural reconstruction of alloy catalysts (e.g., AuCu) may induce metal-metal bond cleavage or phase segregation, altering the electron density of active sites and consequently affecting intermediate stability. Additionally, reconstructed metal-metal interactions further tune intermediate stability. In this study, we found that AuCu alloys

enable the co-adsorption of OH^- on electron-deficient $\text{Cu}^{\delta+}$ sites and glucose on electron-rich $\text{Au}^{\delta-}$ sites, stimulating the formation of oxidative $^*\text{OH}$ species and intermediates. This strengthens the stability of adsorbed intermediates, inhibits C-C bond cleavage, and facilitates end-group oxidation to generate gluconic acid with high selectivity. Certainly, in the electrooxidation of organic molecules such as glucose, C-C bond cleavage requires overcoming a relatively high activation energy barrier. Its occurrence depends on the ability of catalyst active sites to regulate this process. For instance, the orientation of the C-C bond in adsorbed intermediates determines its susceptibility to activation; the presence of highly reactive oxygen species on the catalyst surface may attack C-C bonds; moreover, the stability of intermediates also plays a role—overly stable reaction intermediates (e.g., aldehydes, ketones) on active sites may inhibit C-C bond cleavage, whereas unstable intermediates tend to undergo C-C scission. This represents a complex coordination process, which was not extensively discussed in this manuscript. Future studies will employ in situ characterization to capture the evolution of active sites during reconstruction and combine DFT calculations to verify C-C bond cleavage energy barrier changes, further clarifying the structure-activity relationship.

Q3. Authors said “Previous investigations verified that noble metal catalysts (Au, Pt, and Pd) were widely used for biomass molecules electrooxidation to target product, owing to their moderate C-C bond-breaking capabilities.” This is not true. It has clearly been shown recently by Faverge et al. that Pt and Pd were not selective and favored C-C bond breaking, but that Au was selective at low potentials. (See Faverge et al., ACS Catal. 13 (2023) 2657–2669).

Response: Thank you for pointing out the inaccuracy in our original statement regarding the C-C bond cleavage selectivity of noble metal catalysts. We fully agree with your critical comment and deeply appreciate this correction, as it highlights the need for precise differentiation between the catalytic behaviors of Au, Pt, and Pd in biomass electrooxidation. Upon revisiting the literature, we recognize that our previous generalization overlooked the distinct selectivity trends of these noble metals, and we have revised the description accordingly to align with recent authoritative findings.

As correctly noted by the reviewer, the selectivity of noble metal catalysts for C-C bond cleavage is not uniform across Au, Pt, and Pd. Recent studies by Faverge et al. (ACS Catal. 2023, 13, 2657–2669) have clearly demonstrated that:

Pt and Pd catalysts exhibit poor selectivity for biomass electrooxidation, as their strong oxophilicity and high activity toward C-C bond cleavage favor deep fragmentation into low-value C1 products (e.g., CO₂, formic acid), even at moderate potentials. In contrast, Au catalysts display unique selectivity at low potentials (typically <0.8 V vs RHE), where their moderate adsorption energy for organic intermediates inhibits excessive C-C bond cleavage, enabling the formation of high-value C-C retained products (e.g., gluconic acid from glucose oxidation).

The original sentence has been revised to:

"Previous studies have confirmed that the C-C bond cleavage behavior of noble metal catalysts exhibit significant differences: Pt and Pd are prone to causing deep C-C bond cleavage without selectivity; While Au catalysts, due to their moderate adsorption energy for intermediates, show excellent C-C bond retention selectivity at low potentials (Faverge et al., ACS Catal. 2023, 13, 2657–2669). This characteristic makes Au an ideal catalyst for the electro-oxidation of biomass to produce high-value products." (Page 4)

Q4. Authors said “For alkaline GOR, the adsorbed glucose on the catalyst surface undergoes a dehydrogenation caused by *OH to form an intermediate, and the subsequent C=O activation followed by OH coupling”. This mechanism is not true. It has again been shown recently by Neha et al. (Electrocatalysis 14 (2023) 121-130) and Faverge et al. (See Faverge et al., ACS Catal. 13 (2023) 2657–2669) that the oxidation of glucose could start in a potential range where OH does not adsorb on Au.

Response: We sincerely thank the reviewer for this critical and insightful comment. The reviewer points out the inaccuracy in the mechanistic description in introduction. According to the works of Neha et al. and Faverge et al., the oxidation of glucose could start in a potential range where OH⁻ does not adsorb on Au. The reaction of *OH and glucose only happens from 0.4 V vs. RHE, where the adsorption of OH⁻

starts on the Au surface. In situ Raman (Faverge et al.) confirms the absence of OH⁻ adsorption below 0.4 V, where glucose oxidation initiates via direct dehydrogenation of the C1 aldehyde group, forming enediol intermediates through a rate-limiting electron transfer step (Tafel slope= 120 mV dec⁻¹, Neha et al.). Above 0.4 V, OH⁻ adsorption on Au (verified by CV peaks and in situ IR, Neha et al.) triggers two key steps: (i) accelerated dehydrogenation via OH-induced H-abstraction (Tafel slope decreases to 60 mV dec⁻¹); (ii) C=O activation and OH⁻ coupling to form gluconic acid (Faverge et al.).

Revised manuscript text:

"Glucose electrooxidation on Au proceeds via two potential-dependent pathways: (1) At < 0.4 V vs. RHE, aldehyde group undergoes direct dehydrogenation to enediol intermediates through OH-independent electron transfer (Faverge et al., ACS Catal. 2023); (2) Above 0.4 V vs. RHE, OH⁻ adsorption activates C=O bonds and accelerates dehydrogenation, promoting OH⁻ coupling to gluconic acid (Neha et al., Electrocatalysis 2023)." (Page 4)

Q5. Why chronoamperometry studies are performed at 1.0 V vs RHE and not at 0.7 V vs RHE? 1.0 V vs RHE is a very high potential, moreover, it is possible that the deactivation is less at high potential than at intermediate ones. The deactivation at intermediate potentials, 0.7-0.8 V vs RHE is not due to the formation of surface oxides, but according to Tominaga et al. (Electrochem. Commun. 9 (2007) 1892–1898) to adsorbed linearly gluconic species. Authors have to discuss this aspect.

Response: We thank the Reviewer for this comment. As pointed out by the reviewer and elucidated by Tominaga et al., the primary deactivation pathway at intermediate potentials may occur due to the adsorption of gluconate species. This is distinct from surface oxidation of Au. In response to this comment, we have now performed the chronoamperometry tests at 0.7 V and 1.0 V vs. RHE under a constant glucose concentration (Fig. 2g). The pure Au electrode exhibits a decreasing current density at both 0.7 V and 1.0 V vs. RHE, which is due to the adsorption of gluconate species and surface oxide of Au. By contrast, the current density of Au₄Cu₂ remains stable at both

0.7 V and 1.0 V vs. RHE, demonstrating a high poisoning tolerance during glucose electrooxidation. And we have included corresponding discussion in the revised manuscript:

“In order to eliminate the influence of glucose consumption on the current density, a home-made reactor was designed to keep a constant glucose concentration for glucose electrooxidation by continuous supply of flowing electrolyte (Fig. S18). The current density of Au is continuously decreasing over time at 0.7 V vs. RHE (Fig. 2g). Notably, the current of Au decreases more rapidly as the reaction potential increasing to 1.0 V vs. RHE, suggesting a deactivation of Au at high potential. The quick deactivation at high potential is related to the surface oxide of Au-OH species to AuO_x on Au catalyst. By contrast, the current density of Au₄Cu₂ alloy remains stable at 1.0 V vs. RHE. It is reported that the deactivation of Au catalyst at 0.7 V vs. RHE was caused by the adsorbed linearly gluconic species rather than AuO_x. Au₄Cu₂ alloy also exhibits a maintained performance at 0.7 V vs. RHE, demonstrating a high poisoning tolerance during glucose electrooxidation.” (Page 12)

Fig. 2g. Chronoamperometric tests under a constant glucose concentration.

Q6. Authors said “Fig. 3b describes the representative OH⁻ adsorption and desorption curves. Au₄Cu₂ shows an OH⁻ oxidation adsorption peak with lower onset potential and increased intensity than that of Au.” I think that this explanation is wrong. Indeed, can the authors explain how the desorption of OH⁻ (reduction of the surface) could appear at higher potential than its oxidation (adsorption of OH⁻)? The first oxidation peak is

related with the reduction peak at ca. 0.3 V vs RHE, and is certainly due to the Cu/CuO redox couple!!!

Response: Thank you very much for pointing out this issue. You are right, and the first oxidation peak of CV can be assigned to the oxidation of Cu (*J. Am. Chem. Soc.* 2019, 141, 31, 12192-12196). Hence, it will be very difficult to obtain the correct result of the adsorption of OH⁻ by CV curves. Therefore, we have provided additional data. In situ FTIR spectra of Au₄Cu₂ and Au in 1 M KOH have been performed to investigate the adsorption of OH⁻ (Fig. 3d and 3e). The O-H vibration peak intensity of Au₄Cu₂ is higher than that of Au. And the *OH peak of Au₄Cu₂ can be detected at a lower voltage. Based on the new experimental data, we can conclude that Au₄Cu₂ alloy was credited with the enhanced adsorption capacity for OH⁻.

The corresponding discussion have been included in the revised manuscript:

*“The adsorption activity of OH⁻ was investigated by the in situ FTIR spectra in 1 M KOH (Fig. 3d and 3e). A broad band around 3520 cm⁻¹ is assigned to *OH on Au₄Cu₂, which is detected at 0.3 V vs. RHE. By contrast, the O-H vibration peak of Au emerges at higher potential of 0.4 V vs. RHE. In addition, Au₄Cu₂ exhibits higher O-H vibration peak area than Au at the same potential (Fig. 3f), suggesting that more OH species are formed and accumulated on the surface of Au₄Cu₂ alloy. The above results indicate that alloy with charge polarization sites contributes to the adsorption and activation of OH⁻.” (Page 15)*

Fig. 3. In situ FTIR spectra of (d) Au₄Cu₂ and (e) Au in 1 M KOH for adsorption of OH⁻. (f)

Changes of peak area in of OH⁻ adsorption from in situ FTIR spectra.

Q7. The in situ infrared spectroscopy study does not give any information on the mechanism conversely to what explains the authors. They should refer to specialists in this domain. First, the fingerprint region of organics is between 800 and 2000 cm^{-1} . Here, nothing can be drawn from the IR study. It is very strange that no signal were recorded in the 1200 – 1600 cm^{-1} region. “The vibration peaks at 1680 belonging to carbonyl intermediates (*CO-R) are detected on Au_4Cu_2 ”; This region is very close to the interfacial water vibration band (1640 cm^{-1}) and it is difficult to conclude. “characteristic vibration band of carboxyl intermediates (COO^- , 1535 cm^{-1})” Carboxyl are generally observed around 1580 cm^{-1} , not 1535.

Response: Thanks for the valuable comment. We have referred to specialists for the in situ FTIR study. We have also studied some relevant literature. With the help of specialists, we reconducted the in situ FTIR study test, as shown in Fig. 3b and 3c. The new FTIR results can provide sufficient information on the mechanism.

“In situ Fourier transform infrared spectroscopy (FTIR) was employed to get insights into the mechanism of the glucose electrooxidation over Au_4Cu_2 . The potential-dependent FTIR from 0.1 to 1.2 V vs. RHE provides critical information of the glucose oxidation pathways (Fig. 3b and 3c). Two peaks at 2924 and 2860 cm^{-1} corresponding to C-H stretching vibration emerge on Au_4Cu_2 alloy, associated with glucose adsorption. The peak intensities of C-H stretching over Au_4Cu_2 are more evident than that for Au (Fig. S22), indicating favorable adsorption of glucose for Au_4Cu_2 relative to Au, due to the charge-polarized $\text{Au}^{\delta-}$ sites. The slight attenuation of the glucose adsorption peaks indicates rapid replenishment of consumed glucose (Fig. S23), ensuring a stable GOR process. Importantly, the distinct peaks appear on Au_4Cu_2 at 0.4 V vs. RHE from 1200 to 1800 cm^{-1} , which also increase at an enhanced potential. By contrast, the weaker peaks are detected for Au at 0.5 V vs. RHE. And no IR signal is received on pure Cu electrode at the same potentials (Fig. S24). A peak at 1666 cm^{-1} assigned to vibration band of H_2O appears first before glucose oxidation. As the reaction progresses, the bands at 1577 cm^{-1} are attributed to the asymmetric stretching of COO^- , while the bands at 1410 and 1326 cm^{-1} corresponds to symmetric stretching of COO^- . The different vibration modes of COO^- suggest the oxidation of glucose to gluconate. The

vibration bands at 1755 cm^{-1} belonging to C=O stretching of carbonyl species are detected on Au_4Cu_2 , which is assigned to carbonyl intermediates (*CO-R) for glucose oxidation. *CO-R intermediates can be directly converted into gluconic acid by Au_4Cu_2 alloy, preventing the C-C bond cleavage of glucose and the generation of byproduct.”

(Page 14-15)

Fig. 3 In situ electrochemical FTIR spectra of (b) Au_4Cu_2 and (c) Au at different potentials for GOR in 1 M KOH with 0.2 M glucose

Minor

“Upon alloying Cu with Au, the diffraction peaks of Au_4Cu_2 shift obviously to a higher angle, which is ascribed to the lattice compression. Furthermore, the diffraction peaks shift to higher angle with an increasing amount of Cu, implying the formation of AuCu alloy.” Does it fit with the Vegard's law? And what is the level of alloying? Is it coherent with the other structural characterizations?

Response: Thanks for the valuable comment. The result of lattice compression of Au_4Cu_2 from XRD is coherent with the TEM results (Fig. S4), which show that Au_4Cu_2 alloy exhibits a slightly smaller lattice spacing than that of pure Au. We have also provided the result of concentration dependence of the lattice constant. The XRD results of alloys give a positive deviation from Vegard's law. We also calculated the alloy composition from Vegard's law. Base on the calculated composition, we concluded the level of alloying, 5.85, 15.16 and 21.51% of the Cu remain out of the solid solution alloys in Au_5Cu_1 , Au_4Cu_2 and Au_3Cu_3 , respectively. The corresponding discussion have been included in the revised manuscript:

“The compositions of the alloys derived from the inductively-coupled plasma-optical emission spectrometry (ICP-OES) and Vegard’s law are listed in Table S1. The concentration dependence of the lattice constant for AuCu alloys exhibits a positive deviation from the linear dependence calculated by Vegard’s law (Fig. S8). By simple arithmetic, we conclude that 5.85, 15.16 and 21.51% of the Cu remain out of the solid solution alloys in Au_5Cu_1 , Au_4Cu_2 and Au_3Cu_3 , respectively.” (Page 8)

Fig. S8. Calculated and experimental lattice constants for the AuCu alloy. Calculated lattice constants is obtained from Vegard's law.

“The electron transfer from Bi to its surrounding Au atoms”. Bi ???

Response: We sincerely thank the reviewer for pointing out this critical typo. The sentence should correctly describe as follows: "electron transfer from Cu to its surrounding Au atoms,". We have revised the sentence accurately in the revised manuscript.

It is not clear what are the reactant? OH_{ads} or OH radical?

Response: Thanks for the valuable comment. The reactant is unequivocally the adsorbed hydroxyl species (OH_{ads}) rather than OH radical. We have revised the manuscript throughout to ensure absolute clarity of reactant.

Reviewer #4:

The authors report an AuCu alloy that serves as an efficient electrocatalyst for the selective oxidation of glucose to potassium gluconate at high current density. The alloy enables co-adsorption of OH⁻ on electron-deficient Cu^{δ+} sites and glucose on electron-rich Au^{δ-} sites, promoting the formation of reactive *OH and key intermediates. Preferential OH adsorption on Cu^{δ+} suppresses Au-OH formation and subsequent oxidation to AuO_x, mitigating deactivation. Au₄Cu₂ achieves 97.6% selectivity to potassium gluconate and reaches 500 mA cm⁻² at 0.88 V vs RHE. In a membrane-free flow electrolyzer, Au₄Cu₂ delivers stable operation with a potassium gluconate productivity of 9.46 mmol cm⁻² h⁻¹ and a Faradaic efficiency of 93.6%. However, this manuscript do not meet the requirement for publishing in Nat Comm before addressing the following issues.

Response: Thanks for giving us an opportunity to revise this manuscript, and we have tried our best to revise our manuscript based on the helpful comments from the Reviewer.

Q1. Glucose is listed as C₆H₁₂O₂ (which should be C₆H₁₂O₆). This is a fundamental error in “Chemicals.”

Response: Thank you very much for pointing out this issue. The fundamental error in “Chemicals.” has been revised in the revised manuscript.

Q2. Lattice spacings are reported in nm (e.g., 2.21 nm ... 2.37 nm), clearly meant to be Å.

Response: Thank you very much for pointing out these issues. We have made appropriate corrections in the revised manuscript.

“Fig. 1c is the magnified lattice image of marked region in Fig. 1b, displaying a highly ordered structure with a spacing of 0.221 nm. Au₄Cu₂ alloy exhibits a slightly smaller lattice spacing than that of pure Au (0.237 nm) (Fig. S4)”(Page 7)

Q3. The Faradaic efficiency formula is wrong as written: FE must be FE=($nF \times$ moles of product)/ Q , but the equation shown omits the total charge Q and duplicates F . Also

conflicting text says “C is the total electron transfer”).

Response: Thank you very much for pointing out these issues. We have made appropriate corrections of the Faradaic efficiency formula in the revised manuscript.

“Faraday efficiency (FE) was calculated by the following equation, where Q is the total charge that has passed through the electrode, n (2) is the number of electron transfer for gluconic acid product from glucose, F is the Faraday constant (96485 C mol⁻¹):
(Page 26-27)

$$FE (\%) = \frac{n \times F \times \text{Moles of product}}{Q} \times 100\% \quad (3) "$$

Q4. page 8, the differential discussion suddenly states “electron transfer from Bi to Au”, although the system is Au–Cu; this looks like copy-paste from another study.

Response: We sincerely thank the reviewer for pointing out this critical typo. The sentence should correctly describe as follows: "electron transfer from Cu to its surrounding Au atoms,". We have revised the sentence accurately in the revised manuscript.

Q5. Very high PGA selectivity (up to 97.6%) and FE (95–99.9%) are claimed, yet product analysis relies on UV-HPLC at 210 nm with 5 mM H₂SO₄ on an Aminex HPX-87 column, with no description of sample neutralization from 1 M KOH, matrix effects, lactone/hydrate equilibria, calibration quality, or a full carbon balance.

Response: Thank you very much for pointing out these issues. A supplementary description of sample neutralization from 1 M KOH has been added to the product quantification in Methods. 1 M KOH in liquid product was diluted by 55.6 mM H₂SO₄. In addition, by diluting the sample to an appropriate concentration, the influence of the matrix effect on the measurement results can be reduced. We have added the corresponding discussion in the revised manuscript: *“Before the test, 0.1 mL of the liquid product was added to 0.9 mL of 55.6 mM H₂SO₄ solution for neutralization from 1 M KOH, which could also reduce the matrix effect.”* (Page 26)

And we have provided the external standard calibration curves of potassium gluconate, glucaric acid, and the possible low-carbon intermediate (glyceric acid, lactic acid and

formic acid) in the revised manuscript (Fig. S12). In addition, carbon mass balance during GOR process at 0.7 V vs. RHE is calculated to be 99.7%.

Fig. S12. The HPLC spectra of the standard substances and the corresponding calibration curves established with the external standards.

Q6. The mechanism repeatedly equates the surface *OH needed for GOR with solution •OH radicals detected by DMPO-EPR after electrolysis. DMPO distinguishes free radicals in solution, not the metal-bound *OH adsorbate that governs alkaline alcohol oxidation; the two are not interchangeable evidence.

Response: Thanks for the valuable comment. Considering that *OH can be oxidized to •OH, we intended to evaluate the surface *OH indirectly by the concentration of •OH. However, this is indeed not direct evidence of the metal-bound *OH adsorbate. Hence, we have performed the in situ FTIR spectra of Au₄Cu₂ and Au in 1 M KOH for investigation of OH⁻ adsorption (Fig. 3d and 3e). The O-H vibration peak intensity of Au₄Cu₂ is higher than that of Au. And the *OH peak of Au₄Cu₂ can be detected at a lower voltage. The corresponding discussion have been included in the revised manuscript:

*“The adsorption activity of OH⁻ was investigated by the in situ FTIR spectra in 1 M KOH (Fig. 3d and 3e). A broad band around 3520 cm⁻¹ is assigned to *OH on Au₄Cu₂, which is detected at 0.3 V vs. RHE. By contrast, the O-H vibration peak of Au emerges*

at higher potential of 0.4 V vs. RHE. In addition, Au_4Cu_2 exhibits higher O-H vibration peak area than Au at the same potential (Fig. 3f), suggesting that more OH species are formed and accumulated on the surface of Au_4Cu_2 alloy. The above results indicate that alloy with charge polarization sites contributes to the adsorption and activation of OH^- .” (Page 15)

Fig. 3. In situ FTIR spectra of (d) Au_4Cu_2 and (e) Au in 1 M KOH for adsorption of OH^- . (f) Changes of peak area in of OH^- adsorption from in situ FTIR spectra.

Q7. All high currents are reported per geometric area of 3D Ni foam without ECSA or roughness factor; comparisons with planar controls (Au foil) are therefore not meaningful.

Response: Thanks for the valuable comment. The Au electrode in this manuscript was obtained using the same preparation method as the alloy electrode without addition of $CuSO_4 \cdot 5H_2O$, using nickel foam as substrate. The meaningful and fair comparison in our work is between the AuCu alloy electrode and pure Au electrode control sample (both on nickel foam). In addition, we have estimated ECSA based on the double-layer capacitance (C_{dl}), and the corresponding discussion have been included in the revised manuscript:

“The CV curves in the non-Faradaic potential range was tested with different scan rates (50, 60, 70, 80, 90 and 100 $mV s^{-1}$) from 100 to 200 mV (vs. RHE) to calculate the effective ECSA of the catalyst. Using the following equation to calculate ECSA, where C_s was the specific capacitance of a metallic surface ($40 \mu F cm^{-2}$), C_{dl} was the double-layer capacitance: (Page 25)

$$ECSA = \frac{C_{dl}}{C_S} \quad (1)$$

“The electrochemical surface area (ECSA) was carried out through cyclic voltammetry (CV) with different scan rates (Fig. S10). Double-layer capacitance (C_{dl}) of Au_4Cu_2 is measured to be 6.64 mF cm^{-2} , surpassing 4.94 , 5.65 and 5.79 mF cm^{-2} of Au , Au_5Cu_1 , and Au_3Cu_3 , respectively (Fig. S11).” (Page 10)

Fig. S10. CV curves of (a) Au, (b) Au_5Cu_1 , (c) Au_4Cu_2 and (d) Au_3Cu_3 at different scan rates from 50 to 100 mV s^{-1} in 1 M KOH and 0.2 M glucose .

Fig. S11. The extracted double-layer capacitances (C_{dl}) of different electrodes using a CV method.

Q8. Authors state ****no iR compensation**** anywhere, yet emphasize unusually low voltages.

Response: Thank you for your question. We acknowledge that the electrochemical data presented were obtained without iR compensation.

Q9. The LSVs at 2 mV s^{-1} reaching hundreds of mA cm^{-2} in concentrated alkali are strongly mass-transport limited, so they are not suitable for kinetic claims. should replace with galvanostatic polarization (steady current steps) and hydrodynamically controlled tests.

Response: Thanks for the valuable suggestion. As suggested, we have added experiments of steady current steps to address the mass-transport limited. Quasi-steady-state linear sweep voltammetry (QS-LSV) was carried out to accurately assess the GOR activity, and each point referred to the stabilized current density was recorded at a certain potential from the I-t test (Fig. 2b). In addition, the QS-LSV plots was further normalized through the ECSA to assess the intrinsic activity of GOR. As shown, the Au_4Cu_2 demonstrated better electrocatalytic performance of GOR compared to Au, Au_5Cu_1 , and Au_3Cu_3 electrode. The QS-LSV has been discussed in the revised manuscript:

“To accurately assess the intrinsic activity of GOR, quasi-steady-state linear sweep voltammetry (QS-LSV) normalized through the ECSA is conducted based on the equilibrated current density under a given potential (Fig. 2b), where Au_4Cu_2 demonstrates the best catalytic activity of GOR.” (Page 10)

Fig. 2b ECSA-normalized QS-LSV plots toward GOR, inset is the steady state Tafel slopes.

Since the alloy catalyst is grown on the foam nickel, the bulk catalyst cannot be tested using the rotating disk electrode system for hydrodynamically controlled tests. Instead, in order to accelerate the mass transfer process, we retested the LSV curve under vigorous stirring, as shown in Fig. 2a. The results showed that accelerating the mass transfer process is beneficial for increasing the current density of electrocatalysts.

Fig. 2a. LSV profiles of electrocatalysts for glucose oxidation in 1.0 M KOH and 0.2 M glucose.

Q10. In this paper, the Au onset (~ 0.2 V vs RHE) is far below literature (~ 0.5 V vs RHE; ACS Catal. 2023, DOI: 10.1021/acscatal.2c05871); should reconcile by detailing reference calibration, pH/electrolyte, scan/hydrodynamics, iR compensation, and onset definition, and include side-by-side CVs.

Response: Thank you for your question. In the literature of ACS Catal. 2023, 13, 2657–2669 (DOI: 10.1021/acscatal.2c05871), for gold, “the onset potential (measured at $I_{\text{anodic}} = 3 \mu\text{A}$, a current that significantly exceeds the baseline current in supporting electrolyte) is estimated to be around 0.29 V vs. RHE for glucose oxidation”. This cyclic voltammograms of Au was performed in 0.1 M NaOH aqueous solution in the presence of 0.1 M glucose, with a scan rate of 1 mV s^{-1} . In our work, the Au onset potential was 0.25 V vs. RHE (10 mA cm^{-2}), which was performed in 1 M KOH in the presence of 0.2 M glucose, with a scan rate of 2 mV s^{-1} . The OH^- concentration had an impact on the activity of biomass platform molecules, the current density and the onset potential may increase with the increase of KOH concentration (Angew. Chem. Int. Ed. 2024, 63,

e202409419). Compared with the work of ACS Catal. 2023, 13, 2657–2669, we used higher OH⁻ concentration for GOR, which resulted in a lower onset potential of Au.

[FIGURE REDACTED]

Figure from ACS Catal. 2023, 13, 2657–2669

We also re-calibrate the reference electrode, and confirmed the reference calibration, pH/electrolyte (0.2 M glucose + 1 M KOH, pH = 13.8), scan/hydrodynamics (2 mV s⁻¹), iR compensation (All the electrochemical curves were used without IR compensation), and onset definition (10 mA cm⁻²) in the methods of the revised manuscript. We have added a statement in the revised manuscript:

“The standard experimental calibration at 25 °C was conducted to maintain the accuracy of Ag/AgCl reference. Pt foil was both the working electrode and counter electrode, and 1 M KOH with 0.2 M glucose was selected as the electrolyte, which was saturated with high-purity hydrogen for 30 min. CV was carried out and centered at the open circuit potential with a scan range of ±0.2 V and a scan rate of 1 mV s⁻¹. The calibrated reference electrode potential is determined by taking the average of the two electrode potentials corresponding to the zero point of current density.” (Page 26)

Q11. O–H stretching near 3570 cm⁻¹ in water-alkali is ambiguous; D₂O substitution or isotopic labeling is needed to support assignment to activated OH rather than bulk water structure.

Response: Thanks for the valuable suggestion. We have performed a D-H isotope labeling experiment to support assignment of O-H stretching to activated OH rather than H₂O, as shown in Fig. S25. The assignment of O-H stretching to *OH has been confirmed. We have added the corresponding discussion in the revised manuscript:

“The OH groups on Au₄Cu₂ are further supported by the D-H isotope labeling

experiment (Fig. S25). The IR band of O-D of Au_4Cu_2 is observed at 2560 cm^{-1} in D_2O and KOD solution at 0.8 V vs. RHE. Then the D-H exchange is conducted by replacing the solution with H_2O and KOH through a peristaltic pump. The O-D vibration peak remains after D-H exchange, indicating that the O-D vibration is assigned to $*OD$ species of catalyst surface rather than D_2O . Such D-H exchange experiment confirms the formation of $*OH$ species.” (Page 15)

Fig. S25. H-D exchange FTIR experiment over Au_4Cu_2 .

RESPONSE TO REVIEWERS' COMMENTS (Version 1)

Reviewer #1 (Remarks to the Author)

The authors replied well to my comments and questions. I think the manuscript can be accepted for publication.

Response: We are grateful to the reviewer's effort in reviewing our manuscript and positive comments on our work.

Reviewer #2 (Remarks to the Author)

The manuscript has been improved following the revisions, but several of the responses provided are not fully convincing and leave important mechanistic questions unresolved. The additional data strengthen the case in part, yet ambiguities remain regarding the interpretation of surface stability and adsorption phenomena.

Response: We are very grateful to your professional review and continued attention on our article. We have carefully addressed all concerns in the revised manuscript and followed your suggestions. We hope that the corrections will meet with approval.

- Thank you for providing the ICP OES and stability data. While the bulk composition appears largely unchanged, it is important to note that ICP OES is not surface sensitive, and catalytic activity is governed primarily by the surface. The weak Cu signal observed in the XPS spectra before and also after GOR suggests possible surface depletion. In such a scenario, the apparent stability may not reflect intact AuCu alloying but could instead arise from (i) enhanced surface roughness due to partial Cu dissolution, which increases the electrochemically active surface area (ECSA), or (ii) Au₄Cu₂ alloy covered by an Au rich overlayer, which may alter poisoning tolerance. To clarify the mechanism, additional surface sensitive or operando characterization would be highly valuable—for example, a more precise estimation of surface Cu content by comparing Au/Cu peak ratios in XPS, or complementary electrochemical surface area measurements. Such analyses would help distinguish whether the observed stable performance originates from genuine Au₄Cu₂ surface stability or from restructuring effects. This distinction is crucial, given that the DFT calculations presented assume the

presence of both Cu and Au atoms in the top surface layer.

Response: Thanks for your insightful comments and valuable suggestions. We agree that ICP-OES analysis is insufficient for characterizing surface composition. The possible surface Cu depletion may contribute to either enhanced surface roughness or the formation of an Au-rich overlayer, leading to two alternative explanations for catalyst stability. To directly address these issues and clarify the mechanism, we have conducted a more surface-sensitive characterization to clarify the surface composition before and after the GOR. The key evidence comes from **time-of-flight secondary ion mass spectrometry** (TOF-SIMS, ULVAC-PHI. INC), which is a highly surface-sensitive technique for surface composition.

- (i) As suggested, we performed a precise quantitative analysis of the XPS data. The results show that the Au 4f peak position remains essentially unchanged (Fig. S20), and the characteristic peak for zero-valent Cu is still clearly observed after GOR. This confirms the persistence of metallic Cu at the surface. Quantitative analysis indicates that the surface Au/Cu atomic ratio increased moderately from 1.14 to 1.30 after GOR, suggesting a slight surface change but not a complete loss of Cu.
- (ii) Complementary surface-sensitive characterization: Additional surface-sensitive techniques would be highly valuable. Hence, we performed the TOF-SIMS to accurately assess the surface Au/Cu ratio of Au₄Cu₂ alloy before and after GOR. TOF-SIMS enables elemental analysis of the outermost atomic layer (1–3 nm) of a sample, offering extremely high surface sensitivity and mass resolution. The surface Au/Cu ratio of alloy is 1.10, which changes slightly to 1.19 after GOR (Fig. S21). A little alteration of surface Au/Cu ratio indicates the stability of AuCu surface. This provides direct evidence that significant dissolution of Cu or severe surface depletion does not occur. Such stability of AuCu surface contributes to the poisoning tolerance and stable performance of GOR.

Fig. S21. Surface Au/Cu ratio of Au₄Cu₂ catalyst before and after GOR obtained from TOF-SIMS.

(iii) Ruling out ECSA change as the main cause of stability: To evaluate the possibility that stability stems from the change of ECSA, the C_{dl} of the Au₄Cu₂ catalyst before GOR test have been provided. The results indicate that the change in ECSA after GOR testing is minor (less than 7.5%). Combined with the stable surface composition revealed by TOF-SIMS, we can conclude that the stable performance originates primarily from the preserved intrinsic activity of the Au-Cu surface, not from a substantial increase in surface area.

Fig. S22. The extracted double-layer capacitances (C_{dl}) of Au₄Cu₂ (a) before and (b) after GOR.

In summary, by employing the surface-sensitive analysis (TOF-SIMS), we have obtained direct evidence for the stability of the surface composition. The high and stable performance of the Au₄Cu₂ catalyst during GOR indeed stems from the stability of the Au-Cu surface alloy, consistent with our DFT model which incorporates both Au and Cu atoms in the active surface layer. We have included corresponding content in the revised manuscript:

“Besides, a systematic study was carried out on the catalyst changes during GOR.

The morphology and composition of Au₄Cu₂ alloy are maintained after a prolonged reaction (Fig. S19 and Table S1). XPS analyses reveal that valence states of Au in alloy remain essentially unchanged (Fig. S20), meanwhile, zero-valent Cu is still observed on the alloy surface after GOR. And Au/Cu atomic ratio from XPS analysis increased from 1.14 to 1.30 after GOR. In order to accurately assess the changes in the metal composition during the reaction, we performed the time-of-flight secondary ion mass spectrometry (TOF-SIMS) for analysis of the outermost atomic layer of a surface. The surface Au/Cu ratio of alloy is 1.10, which changes slightly to 1.19 after GOR (Fig. S21). A little alteration of surface Au/Cu ratio indicates the stability of AuCu surface. In addition, to evaluate the possibility that stability stems from the change of ECSA, we monitored the C_{dl} of Au₄Cu₂ before and after GOR (Fig. S22). The surface Au/Cu atomic ratio increased moderately from 6.16 to 6.64 mF cm⁻² after GOR. Combined with the stable surface composition, it is deduced that the catalytic stability of Au₄Cu₂ originates primarily from the preserved intrinsic activity of the AuCu surface.” (Page 12-13)

- The surface enhancement for Au₄Cu₂ is confirmed by the authors: “Double-layer capacitance (C_{dl}) of Au₄Cu₂ is measured to be 6.64 mF cm⁻², surpassing 4.94, 5.65 and 5.79 mF cm⁻² of Au, Au₅Cu₁, and Au₃Cu₃, respectively (Fig. S11).” In Fig. 2b, once normalized by ECSA, the activity of Au₄Cu₂ at low overpotential appears close to that of pure Au or other AuCu alloys. The difference in activity is observed above 0.5 V, where the current of Au₄Cu₂ continues to increase with potential.

Response: Thanks for your valuable comments. Glucose electrooxidation on Au proceeds via two potential-dependent pathways: (1) At < 0.4 V vs. RHE, aldehyde group undergoes direct dehydrogenation to adsorption intermediates through OH-independent electron transfer (C₆H₁₂O₆ + 2Au → Au-C₆H₁₁O₆ + Au-H); (2) Above 0.4 V vs. RHE, OH⁻ adsorption activates C=O bonds and accelerates dehydrogenation, promoting OH⁻ coupling to gluconic acid (*ACS Catal.* 2023, 13, 2657-2666; *Electrocatalysis* 2022, 14, 121-130). Since -OH species are generated at a potential of 0.3 V and 0.4 V for Au₄Cu₂ and Au, respectively. We believe that the glucose oxidation reactions undergo a OH-independent process < 0.4 V vs. RHE for Au, Au₅Cu₁ and

Au₄Cu₂. At this stage, the main active site is Au, hence, there is no difference in activity < 0.4 V for these three samples. The difference in activity is observed above 0.5 V, when the OH⁻ coupling participates in the glucose oxidation reaction. Due to the strong charge polarization of alloy enables OH⁻ adsorbed on electron-deficient Cu^{δ+} sites and glucose adsorbed on electron-rich Au^{δ-} sites, the current of Au₄Cu₂ continues to increase with potential.

- The in situ FTIR data are helpful, but it should be noted that Cu dissolution and surface restructuring could also modify the OH vibrational region. As a result, the observed differences between Au₄Cu₂ and Au may not solely reflect enhanced OH⁻ adsorption, but could partly arise from surface modification effects.

Response: We thank the Reviewer for this comment. We fully agree that Cu dissolution and surface restructuring could indeed influence the vibrational region of OH. However, additional experiments suggest that the primary origin of the OH vibrational change is likely related to enhanced OH⁻ adsorption. Our conclusion is based on the following three points:

- (i) Our supplementary surface-sensitive characterization (specifically, TOF-SIMS) indicates that the Au-Cu surface alloy composition remains during GOR. The Cu dissolution or surface restructuring appears to be minimal.
- (ii) The more negative onset potential for GOR on Au₄Cu₂ versus pure Au, which is consistent with a lower thermodynamic barrier for the initial deprotonation step of glucose facilitated by OH⁻. In addition, DFT calculations predict a higher affinity for OH⁻ on the Au₄Cu₂ surface compared to pure Au.
- (iii) To further clarify that the change of OH vibrational region primary stems from the enhanced OH⁻ adsorption rather than surface restructuring. We have performed Raman spectra of the Au₄Cu₂ alloy in 1 M KOH. The spectra also show bands at 532 and 683 cm⁻¹ at 0.3 V vs. RHE, which is same as the result of in situ FTIR spectra. According to the reference (*J. Am. Chem. Soc.* 2019, 141, 12192-12196), the 683 cm⁻¹ band belongs to the bending mode of Cu-OH. 532 cm⁻¹ band belongs to a top site OH stretching mode, where OH participates in hydrogen bonding with

another OH. Therefore, these two peaks belong to the OH⁻ adsorption. In addition, there are no signals of Cu₂O and CuO that originate from surface restructuring. We have included corresponding content in the revised manuscript (**Page 16**).

In situ Raman spectra of the Au₄Cu₂ alloy in 1 M KOH.

Reviewer #4 (Remarks to the Author)

I think the authors have mostly addressed the issues.

Response: We are grateful to the reviewer's effort in reviewing our manuscript and positive comments on our work.